# Indigenous Uses, Phytochemical Analysis, and Anti-Inflammatory Properties of Australian Tropical Medicinal Plants

**DOI:** 10.3390/molecules27123849

**Published:** 2022-06-15

**Authors:** Karma Yeshi, Gerry Turpin, Tenzin Jamtsho, Phurpa Wangchuk

**Affiliations:** 1Centre for Molecular Therapeutics, Australian Institute of Tropical Health and Medicine, James Cook University, Building E4, McGregor Rd, Smithfield, Cairns, QLD 4878, Australia; phurpa.wangchuk@jcu.edu.au; 2Tropical Herbarium of Australia, James Cook University, Building E1, McGregor Rd, Smithfield, Cairns, QLD 4878, Australia; gerry.turpin@des.qld.gov.au; 3Yangchenphug High School, Ministry of Education, Thimphu 11001, Bhutan; jamtshooo@gmail.com

**Keywords:** anti-inflammatory, medicinal plants, tropical, aboriginal people, inflammation, inflammatory, phytochemistry

## Abstract

Australian tropical plants have been a rich source of food (bush food) and medicine to the first Australians (Aboriginal people), who are believed to have lived for more than 50,000 years. Plants such as spreading sneezeweed (*Centipeda minima*), goat’s foot (*Ipomoea pes-caprae*), and hop bush (*Dodonaea viscosa* and *D. polyandra*) are a few popular Aboriginal medicinal plants. Thus far, more than 900 medicinal plants have been recorded in the tropical region alone, and many of them are associated with diverse ethnomedicinal uses that belong to the traditional owners of Aboriginal people. In our effort to find anti-inflammatory lead compounds in collaboration with Aboriginal communities from their medicinal plants, we reviewed 78 medicinal plants used against various inflammation and inflammatory-related conditions by Aboriginal people. Out of those 78 species, we have included only 45 species whose crude extracts or isolated pure compounds showed anti-inflammatory properties. Upon investigating compounds isolated from 40 species (for five species, only crude extracts were studied), 83 compounds were associated with various anti-inflammatory properties. Alphitolic acid, Betulinic acid, Malabaric acid, and Hispidulin reduced proinflammatory cytokines and cyclooxygenase enzymes (COX-1 and 2) with IC_50_ values ranging from 11.5 to 46.9 uM. Other promising anti-inflammatory compounds are Brevilin A (from *Centipeda minima*), Eupalestin, and 5′-methoxy nobiletin (from *Ageratum conyzoides*), Calophyllolide (from *Calophyllum inophyllum*), and Brusatol (from *Brucea javanica*). *D. polyandra* is one example of an Aboriginal medicinal plant from which a novel anti-inflammatory benzoyl ester clerodane diterpenoid compound was obtained (compound name not disclosed), and it is in the development of topical medicines for inflammatory skin diseases. Medicinal plants in the tropics and those associated with indigenous knowledge of Aboriginal people could be a potential alternative source of novel anti-inflammatory therapeutics.

## 1. Introduction

Since time immemorial, plants have been a vital source of food, shelter, clothing, tools, and weapons for humankind. Before modern allopathic medicines, early civilizations dealt with illnesses and diseases mostly with natural products from native plants and fungi, and they were taken either in raw or partially processed form. Moreover, these plants have been one of the vital sources of modern drugs, and medicinal plants still play a significant role in the biodiscovery of chemical leads for developing novel therapeutics. Of 52,885 medicinal plants identified globally [1], the phytochemical profile of only about 15% of these species has been reported thus far [2]. The World Health Organization (WHO) estimated that about 80% of the population in developing countries still rely on medicinal plants for their primary healthcare [3,4]. Many indigenous communities globally, including Australian indigenous people—popularly known as Aboriginal people, still use plants for preventive and curative purposes, despite having adequate access to advanced modern healthcare systems.

The Australian Aboriginal people are known to have occupied the country more than 50,000 years ago, and currently, they constitute 3.3% of the total Australian population [5,6]. Aboriginal people have developed a profound connection with their native flora and fauna. Their longstanding survival could have resulted from the prolonged use of medicinal plants in their diet and home remedies [7], and they still use medicinal plants in their day-to-day life. However, as they have already merged with mainstream modern society, it has become crucial to properly document their vast indigenous knowledge for their future generation [8]. The Northern Territory government, in collaboration with the Commonwealth in the 1980s, compiled “Traditional Bush Medicines,” an Aboriginal pharmacopeia of the Northern Territory [9], a first-ever initiative to record dying Aboriginal medicinal lore. Since then, more collaborations have occurred between Aboriginal communities and scientists from various universities across Australia to explore Aboriginal medicinal plants [10,11]. Exploring indigenous food and medicinal plants may give a wealth of potential candidates for novel therapeutics, and Australian native plants could be an intriguing source. The geographic isolation of Australia from Gondwana and other parts of the world for over 65 million years [12,13] has become home to unique and complex flora, where approximately 85% of its vascular plants are endemic species [10]. Plants growing in the tropics produce more phenolics, flavonoids, and terpenoids during adaptation to its extreme vegetative and climatic conditions [14]. Phenolics and flavonoids are antioxidative and anti-inflammatory [15,16,17], and thus tropical plants may yield novel drug leads for treating infectious and non-infectious diseases, including chronic inflammatory conditions [18]. More than 900 medicinal plant species have been recorded in the Tropical region of Australia (shaded green in Figure 1) [19]. Aboriginal people have used these medicinal plants for treating various diseases, including inflammations and inflammatory-related diseases.

Inflammation is usually characterized by swelling and redness, pain, high body temperature (or heat), and loss of tissue function triggered by pathogens, toxic compounds, and damaged cells, whereby our immune system usually responds to initiate the healing process [21]. Anti-inflammatory drugs are used to manage inflammation (redness and swelling) and pain when conditions become severe. Currently, only FDA (United States Food and Drug Administration)-approved nonsteroidal anti-inflammatory drugs (NSAIDs) such as ibuprofen, indomethacin, naproxen, cyclooxygenase-2 enzyme (COX-2) (categorized as non-selective NSAIDs), and celecoxib (selective NSAIDs), are available to manage inflammations and inflammatory processes [22]. These NSAIDs are associated with adverse effects on gastrointestinal, cardiovascular, nervous, hepatic, respiratory, renal, and hematologic systems [23]. Therefore, there is a need for safer alternative plant-based anti-inflammatory drugs. This study comprehensively reviewed 45 medicinal plants of tropical Australia used by Aboriginal people for inflammation and inflammatory-related diseases and explored their potential for biodiscovery of promising drug candidates to develop anti-inflammatory therapeutics. These medicinal plants also grow in Australia’s sub-tropical and rainforest eco-regions.

## 2. Plant Selection and Literature Review Methods

The anti-inflammatory medicinal plants included in this review were selected following the schematic method outlined in Figure 2. Initially, we have freely listed medicinal plants which are used by Aboriginal people of Australia, referring to various books on Australian medicinal plants [8,24,25,26,27,28], bush food [29,30,31], and bush medicines [27,32,33,34], followed by further verification by the senior Aboriginal Ethnobotanist at Traditional Indigenous Ethnobotany Centre, Australian Tropical Herbarium, James Cook University. Subsequently, we selected those medicinal plants found in the tropical regions of Australia (175 species) by cross-checking their distribution on the Australian virtual herbarium site [35], as medicinal plant use records are known to be more prevalent in the northern regions of Australia, where traditional knowledge is much preserved [36,37,38]. After that, we selected only those medicinal plants (78 species) primarily associated with the treatments of inflammation and inflammatory-related diseases through data mining using the online database (GoogleScholar and PubMed). We used keywords such as ‘anti-inflammatory’, ‘anti-inflammatory uses’, ‘inflammation,’ ‘inflammatory’, ‘pro-inflammatory cytokines’, ‘inhibition’, ‘anti-inflammatory compound’, and the name of each medicinal plant to search for the relevant literature and the information. Once the list of anti-inflammatory medicinal plants was confirmed, we reviewed their phytochemical and pharmacological properties (crude extracts and pure isolated compounds) and selected 45 species by searching the published literature indexed in PubMed using similar search keywords mentioned above. We have included only those research articles that were indexed in PubMed and Scopus databases and excluded those not indexed in these databases since the quality of the information was questionable and unreliable. Most medicinal plants included in this review are also reported to grow in the tropics of other countries, which are widely studied. However, there are limited phytochemical and pharmacological studies on medicinal plants endemic to tropical Australia. All chemical structures included in this review are referred from PubChem [39] and ChemSpider [40].

## 3. Ethnomedical Uses of Selected Medicinal Plants

Out of 78 tropical medicinal plants used by the Aboriginal people of Australia for treating inflammation and inflammatory-related diseases, 45 species were selected for this review (Table 1). Most of these medicinal plants were trees (18 plant species), followed by herbs (16 species) and shrubs (10 species), and one grass (Figure 3a). All parts of the plants were indicated for at least one or more inflammatory conditions (Table 1). Aerial parts such as leaves and bark were used maximum, followed by roots and wood. Medicinal plant exudates such as gums, latex, and sap were indicated least used. Fruits are known to be rich in antioxidant compounds such as phenolics and flavonoids [41], but only fruits of two medicinal plants were found to be used Figure 3b).

A paste made from different parts of medicinal plants is applied to cuts, wounds, and sores to promote healing and relieve body and joint pains. A few medicinal plants are used in combination with other plants and products. For example, whole plant part of *Centipeda minima* is mixed with *C. cunninghamii* and *C. thespidioides*, to prepare decoction or infusion to wash inflamed eyes due to the conjunctiva and purulent ophthalmia [42,43]. Powdered leaves of *Centella asiatica* are mixed with lime to treat sores in babies [42]. Some of these medicinal plants are also used in other countries for treating similar types of diseases. For example, Aboriginal people use the mashed plant of *Ageratum conyzoides* to enhance wound healing [32,42], and Nigerians use it for treating wounds [32]. The powdered nut kernel of *Calophyllum inophyllum* is mixed with red pigment and water, which is then rubbed on the affected body parts to ease body pain [42]. This plant is used for treating rheumatism in India [31,32].

Decoctions (prepared from wood and rots) and infusions (prepared from aerial parts such as leaves and flowers) from at least 20 medicinal plants (Table 1) are either used as a mouth wash for mouth inflammations and toothache or taken orally to treat inflammatory-related conditions, such as dysentery, diarrhoea, bowel problems, sore belly, and colic warts. Besides their benefits, medicinal plants may also contain toxins, such as alkaloids and glycosides, that can cause hepatic and renal failures when consumed directly without being correctly processed. Long-term use of medicinal plants such as *Aconitum* species, *Eucalyptus* species, *Datura* species, *Ephedra* species, and *Lobelia* species can have strong toxicity, mainly in children [44,45]. Aboriginal people are known to possess substantial knowledge about the detoxification of medicinal plants and their antidote therapy [46,47], but it still deserves greater attention.

**Table 1 molecules-27-03849-t001:** Ethnomedical uses and the compounds isolated from Aboriginal tropical medicinal plants of Australia.

Species and Family	Ethnomedical Uses	Countries from Where the Plant Has Been Collected for Chemical Studies	Parts Used for Chemical Isolation	Isolated Compounds
*Acalypha wilkesiana* Müll.Arg. (Euphorbiaceae)	Pulped shoots (i.e., collected when leaves are still red) are applied to cuts and open sores [42].	Nigeria	Leaves; stem and root barks	Gallic acid, Corilagin, Geraniin, Rutin, Kaempferol 3-*O*-rutinoside [48].
*Ageratum conyzoides* (L.) L. (Asteraceae)	Meshed whole plant applied to wounds to enhance healing [32,42].	Brazil, India	Whole plant	5,6,7,8,3′,4′,5′-Heptamethoxyflavone, Coumarin [49]; Ageconyflavones A-C, Linderoflavone B, Eupalestin, Nobiletin, 5,6,7,5′-Tetramethoxy-3′,4′-methylenedioxyflavone, Sinensetin, 5,6,7,3′,4′,5′-Hexamethoxyflavone, 5,6,7,8,3′-Pentamethoxy-4′-hydroxyflavone, 5,6,7,8,3′,5′-Hexamethoxy-4′-hydroxyflavone [49,50].
*Alphitonia excelsa* (Fenzl) Reissek ex Benth. (Rhamnaceae)	Leaves are applied to sore eyes; warm aqueous leaves infusion is used as a bath to ease headaches; decoction from bark, wood, and roots is applied externally to relieve body pains; bark and wood decoction are used as a mouth wash to relieve toothache [34,42].	Philippines	Twigs	Betulinic acid [51].
*Alphitonia petriei* Braid & C.T.White (Rhamnaceae)	A decoction made from the bark is applied externally to relieve body pain [34].	Australia	Leaves; stems	Embolic acid, Alphitolic acid, *trans*- and *cis*-Coumaroyl esters of alphitolic acid, Betulinic acid [52].
*Angophora costata* (Gaertn.) Hochr. ex Britten (Myrtaceae)	An aqueous solution of reddish exudate from the trunk is taken orally against diarrhoea [8,53].	Australia	Leaves	Costatamins A-C [54].
*Antidesma bunius* (L.) Spreng. (Phyllanthaceae)	Indicated for headaches, colds, and fevers [32].	Vietnam	Leaves; fruits	Antidesoside, Podocarpusflavone A, Amentoflavone, Byzantionoside B, Roseoside [55].
*Barringtonia racemosa* (L.) Spreng. (Lecythidaceae)	Pulverized roots are applied to skin sores [42].	Bangladesh, China, India, Taiwan, and Vietnam	Stem bark; seeds; roots; leaves	Olean-18-*en*-3β-*O*-*E*-coumaroyl ester, Olean-18-*en*-3β-*O*-*Z*-coumaroyl ester, Germanicol, Germanicone, Betulinic acid, Lupeol, Taraxerol [56]; 3,3’-Dimethoxy ellagic acid, Dihydromyticetin, Gallic acid, Bartogenic acid, Stigmasterol [57,58]; Rutin [59,60]; Nasimalun A and B [61]; Barringtin D1-D3, and M1, Casuarictin, Tellimagrandin I, Valoneic acid dilactone, Schimawalin A [62]; Isoracemosol A, Racemosaceramide A, Racemosol A and E [58,63]; Barringtogenol C [58]; 3β-*p*-*E*-Coumaroymaslinic acid, *cis*-Careaborin, Careaborin, Maslinic acid, 2α,3β,19α-Trihydroxyolean-12-*ene*-24,28-dioic acid, 3β-*p*-*Z*-coumaroylcorosolic acid, Corosolic acid, 1α,2α,3β,19α-Tetrahydroxyurs-12-*en*-28-oic acid, 19α-Hydroxyl ursolic acid, 3α,19α-Dihydroxyurs-12-*en*-24,28-dioic acid, Tormentic acid, 3-Hydroxy-7,22-*dien*-ergosterol [64]; Barringosides G-I [65].
*Brasenia schreberi* J.F.Gmel. (Combretaceae)	Astringent leaves are used for dysentery [31,42].	Canada		Quercetin-7-*O*-glucoside, Gallic acid [66].
*Brucea javanica* (L.) Merr. (Simaroubaceae)	Roots and leaves are used as analgesics [32].	China and Thailand	Aerial; seeds; roots	Brusatol [67]; Demethyl-dehydrobrusatol, Deacetyl-yadanzioside I, Javanicoside G, Yadanziolide C and E, Bruceine A-D and H, Bruceoside A-E, Yadanzioside C and I, Yadanzioside K and L, Dehydrobruceine B, Dehydro-bruceantinol, Deacetylated isobrucein B [68]; brujavanol A and B, bruceine, 11-dehydroklaineanone, 15β-hydroxyklaineanone, 14,15β-dihydroxyklaineanone, 15β-*O*-acetyl-14hydroxyklaieanone [69]
*Calophyllum inophyllum* L. (Calophyllaceae)	Nut kernel ground with red pigment is mixed with water and rubbed to ease body pain [42].	China, France, Fiji, French Polynesia, India, Indonesia, Malaysia, Thailand, Taiwan, and Vietnam	Leaves; seeds; twigs; stems; roots	Inophinnin, Inophinone [70,71]; Inophyllin A, Friedelin, Stigmasterol [71,72,73]; Macluraxanthone, Pyranojacareubin, 4-Hydroxyxanthone, Betulinic acid, Inophyxanthone A, Pancixanthone A, Gerontoxanthone B, Jacareubin [71,74,75,76]; Inocalophyllin A and B [77]; Caloxanthone O and P [78]; Tamanolide, Tamanolide D, E1, E2, and P [79,80]; Calophyllolide [81,82]; 3β,23-*epoxy*-Friedelan-28-oic acid, Epifriedelanol, Canophyllal, Canophyllol, Canophyllic acid, 3-oxo-Friedelan-28-oic acid, Oleanolic acid, 3,4-Secofriedelan-3,28-dioic acid, 27-Hydroxyacetate canophyllic acid, 3-oxo-27-Hydroxyacetate friedelan-28-oic acid [73,83,84]; Caloxanthone Q, 2-Deprenylrheediaxanthone B, 6-Deoxyjacareubin [75,85]; 1,3,6,7-Tetrahydroxy-5-methoxy-4-(1′,1′-dimethyl-2′-propenyl)-8-(3″,3″-dimethyl-2″-propenyl)-xanthone, (2′*S*)-7-Hydroxycaloxanthone B, Caloxanthone A-C, 7-Prenyljacareubin, Daphnifolin, Tovopyrifolin C, 1,3,5-Trihydroxyxanthone, 2-Hydroxyxanthone [76]; Inophyllums G-1, G-2, and P [86]; Isocalophyllic acid, Amentoflavone [84,87]; 27-[(*E*)-*p*-Coumaroyloxy]canophyllic acid, 27-[(Z)-*p*-coumaroyloxy]canophyllic acid, Methyl shikimate, (3*S*,5*R*,6*R*,7*E*,9*R*)-3,5,6-Trihydroxy-β-ionyl-3-*O*-β-d-glucopyranoside, Benzyl-*O*-α-l-rhanmopyranosyl (1→6)-β-d-glucopyranoside, Hexylrutinoside, Kaempferol-3-*O*-α-l-rhamnoside, 27-[(*Z*)-*p*-Coumaroyloxy]friedelin-28-carboxylic acid, (22*E*,24*R*)-24-Methyl-5α-cholesta-7,22-diene-3β,5,6β-triol, 3-oxo-Friedelan-28-oicacid [87]; *trans*-2-[2-(Trifluoromethyl)phenyl]-10b,10c-dimethyl-10b,10c-dihydropyrene, *anti*-4-aza-B-Homo-5α-cholestane-3-one [88].
*Centella asiatica* (L.) Urb. (Apiaceae)	Juice derived from the plant is taken orally or applied locally for non-specific ulcerations. Powered leaves mixed with lime are applied to sores on babies, and the plant is also indicated for skin diseases [31,32,42,89].	China, Japan, India, Madagascar, USA, and Vietnam	Whole plant	Asiaticoside, Asiaticoside C, F, G-I, 23-*O*-Acetyl madecassoside, Asiatic acid, Madecassic acid, Madecassoside, 23-*O*-Acetylasiaticoside B, Stigmasterol 3-*O*-β-glucoside, Quercetin 3-*O*-glucuronide [90,91,92,93,94,95]; Inositol, Centellose [92]; 4′-Hydroxyl-7-methoxyl-6-prenyl-3-*O*-*trans*-*p*-Coumaroyl-flavonol, (2*R*,3*R,*2′′*S*)-3-Furanoyl-brosimacutin E, Epigallocatechin 3-*O*-*p*-coumaroate, Pinobanksin-3-propanoate, Kaempferol, Pachypodol, Coryaurone A [94,96]; Asiaticoside B [93,97]; Isomadecassoside [98]; Quadranoside IV, Quercetin, Astragalin, Isoquercetrin [94]; Centelloside E-G, 11-oxo-Asiaticoside B, 11-oxo-Madecassoside, 11(β)-Methoxy asiaticoside B, 11(β)-Methoxy madecassoside, Centellasaponin A, Isoasiaticoside, Scheffoleoside A [93]; 2α,3β,20,23-Tetrahydroxyurs-28-oic acid [99]; Ursolic acid lactone, Ursolic acid, Pomolic acid, Epi-maslinic acid, Corosolic acid, Rosmarinic acid [95].
*Centipeda minima* (L.) A.Braun & Asch. (Asteraceae)	Infusion and decoction from the whole plant, along with other two species (*C. cunninghamii* and *C. thespidioides*) is used to wash eye inflammation due to conjunctiva and purulent ophthalmia [42,43].	China, Japan, Nepal, South Korea, and Thailand	Whole plant	Brevilin A [100,101]; Apigenin, Quercetin-3-Me-ether, Quercetin-3,3′-diMe-ether, Quercetin-3,7,3′-trimethyl-ether, Quercetin-3,7,3′,4′-tetramethyl-ether, Isobutyroylplenolin, Senecioylplenolin, Aurantiamide acetate, Tetrahydrohelenalin, α-Cyperone [102]; 6-*O*-Methylacrylylplenolin, 6-*O*-Isobutyroylplenolin, 6-*O*-Angeloylplenolin [103]; 2β-(Isobutyryloxy)florilenalin [104]; 2*R*,3*R*)-(+)-7,4′-di-*O*-Methyldihydrokaempferol, Iristectorin A, 4′,5,8-Trihydroxy-7-methoxyisoflavone, 3-Trimethoxyquercetin, 3-*O*-Caffeoyl-α-glueopyranose, 3-*O*-Caffeoyl-β-glucopyranose, Quercetin, Epipinoresinol, Hispidulin [105]; Minimaoside A and B [106]; Minimolides G and H [107]; Minimolide A-F, J-L, Cenminolide A, B, Centiplide A, (1*S*,2*S*,4*R*,5*S*,7*R*,8*S*,10*R*)-2α-Tigloyloxy-4α-angeloyloxyguaia- 11(13)-en-8α,12-olide, Centiplide C-I [101,108,109]; 8,10-Dihydroxy-9(2)-methylbutyryloxythymol, 10-Hydroxy-8,9-dioxyisopropylidene-thymol, 8,9,10-Trihydroxythymol, Thymol-β-glucopyranoside, 9-Hydroxythymol, 8,10-Dihydroxy-9-isobutyryloxythymol, 8-Hydroxy-9,10-diisobutyryloxythymol [110]; 4,5β-Dihydroxy-2β-(isobutyryloxy)-10βH-guai-11(13)-en-12,8β-olide, 4-Hydroxyguaia-9,11(13)-dien-12,8β-olide, 2β-(Isobutyryloxy) florilenalin, Pulchellin-2α-*O*-tiglate, Florilenalin-2α-*O*-tiglate [111]; Microhelenalin B and C, Arnicolides B-D, Helenalin-angelate, Helenalin-isovalerate, Helenalin-isobutyrate, Helenalin-3-methyl-2-butanoate, Minimolide E, Minimolide B, 2α-Methoxy-6α-angeloyl-2,3-helenalin [101]; Caloinophyllin A, Nobiletin, Quercetin pentamethyl ether, 3′,4′5,7-Tetramethoxyflavone, 4′,5,7-Trimethoxyflavone, 1,5-Dihydroxyxanthone, 1,8-Dimethoxy-2-hydroxyxanthone, 1,6-Dihydroxy-7-methoxyxanthone, 4-Methoxycaffeic acid [112].
*Cleome viscosa* L. (Cleomaceae)	The whole meshed plant is applied externally to relieve rheumatism, swellings, headaches, colds, ulcers, and open-sores; seeds are eaten to relieve fever and diarrhoea [8,42].	India, USA, Nigeria, and Vietnam	Seeds; aerial; leaves	Quercetin 3-*O*-(2″-acetyl)-glucoside [113]; Malabaric acid, Stigmast-4-en-3-one, Stigmast-4-*ene*-3,6-dione [114]; Cleomaldeic acid [115]; Lupeol [116]; Astragalin, Visconoside A-C, Vincetoxicoside A and B, Kaempferitrin, Kaempferide 3-*O*-β-d-glucopyranoside 7-*O*-α-l-rhamnopyranoside, Kaempferol 3-*O*-β-d-glucopyranoside 7-*O*-α-l-rhamnopyranoside, Isorhamnetin 3-*O*-β-d-glucopyranoside [117,118]; Lactam nonanoic acid [119].
*Clerodendrum inerme* (L.) Gaertn. (Heliotropiaceae)	Crushed leaves and bark are applied on sores [32,42].	China, Egypt, India, Taiwan, Thailand, and Vietnam	Aerial; flowers; roots; leaves	3-Hydroxy-3′,4′-dimethoxychalcone, 3,2′-Dihydroxy-3′,4′-dimethoxychalcone, 5-Hydroxy-7,8-dimethoxyflavone, Eucalyptin [120]; 2-(3-Methoxy-4-hydroxylphenyl) ethyl-*O*-2”,3”-diacetyl-α-l-rhamnopyranosyl-(1→3)-4-*O*-(E)-feruloyl-β-d-glucopyranoside, monomelittoside, Melittoside, Inerminoside A1, Acteoside, Isoacteoside, Campneoside I [121,122,123]; 4α-Methyl-24β-ethyl-5α-cholesta-14,25-dien-3β-ol; 24β-Ethylcholesta-5,9(11),22*E*-trien-3β-ol; 11-Pentacosanone; 6-Nonacosanone, Clerodermic acid [124]; Inerminoside A-D [125,126]; Sammangaosides A-C, Leucosceptoside A, Decaffeoyl-acteoside, Darendoside B, Monomelittoside, Melittoside, (7*S*,8*R*)-Dehydrodiconiferyl alcohol 9-*O*-β-glucopyranoside, (7*S*,8*R*)-Dehydrodiconiferyl alcohol 4-*O*-β-glucopyranoside, β-Glucopyranoside, β-(2′-*O*-β-Xylopyranosyl) glucopyranoside, Salidroside, (*Z*)-3-Hexenyl-β-glucopyranoside, 2,6-Dimethoxy-*p*-hydroquinone 1-*O*-β-glucopyranoside, Seguinoside K [123]; Lup-1,5,20(29)-trien-3-*O*-β-d-glucopyranoside [122]; Octacosane, Friedelin, β-Amyrin [127]; Crolerodendrum A and B, Uncinatone, Harwickiic acid, Acacetin, Kaempferol 3,7,4′-trimethyl ether, 5α,8α-Epidioxyergosta-6,22-diene-3β-ol [128,129]; Inermes A and B, 14,15-Dihydro-15β-methoxy-3-epicaryoptin [130]; Hispidulin, Diosmetin [129].
*Corymbia terminalis* (F.Muell.) K.D.Hill & L.A.S.Johnson (Myrtaceae)	The plant is used for dysentery [131].	Australia	Gum	Cianidanol, Taxifolin, Aromadendrin, Farrerol [132].
*Crinum pedunculatum* R.Br. (Amaryllidaceae)	Crushed whole plant-rubbed on body parts stung by marine organism [32,42].	NA	NA	NA
*Dodonaea polyandra* Merr. & L.M.Perry (Sapindaceae)	The plant is used for toothache, mouth inflammation, cuts, and open wounds [32].	Australia	Leaves; stems; leaf resins	Polyandric acid A [133]; 13,17-Epoxy-13-methyl-15-oxo-labda-7-ene, 17-Hydroxy-13-methyl-labda-7,13*Z*-diene-15-oic acid, 13-Methyl-17-oxo-labda-7,13*Z*-diene-15-oic acid, Labdane [134]; 15,16-Epoxy-8α-(benzoyloxy)methylcleroda-3,13(16),14-trien-18-oic acid, 15,16-Epoxy-8α-(benzoyloxy)methyl-2α-hydroxycleroda-3,13(16),14-trien-18-oic acid, 15,16-Epoxy-8α-(benzoyloxy)methyl-2-oxocleroda-3,13(16),14-trien-18-oic acid, 15,16-Epoxy-2α-benzoyloxycleroda-3,13(16),14-trien-18-oic acid [135]; 5,7,4′-Trihydroxy-3′(3-methylbut-2-enyl)-3-methoxy flavone, 5,7-Dihydroxy-3′(3-methylbut-2-enyl)-3,4′-dimethoxy flavone, 5,7,4′-Trihydroxy-3′,5′(3-methylbut-2-enyl)-3-methoxy flavone, 5,7,4′-Trihydroxy-3′,5′(3-methylbut-2-enyl)-3,6-dimethoxy flavone, Viscosol, 5,4′-Dihydroxy-3,7-dimethoxyflavone [136].
*Dodonaea viscosa* (L.) Jacq. (Sapindaceae)	Leaves are chewed to relieve toothache; root juice is used as a mouthwash; leaf juice is used to heal stonefish and stingray wounds; root decoction is applied to wounds [34,42].	Cameroon, China, and Mexico	Stems; bark	Dodovisins A-F, Dodovisnoid E, (+)-hardwickiic acid, *ent*-15,16-Epoxy-1,3,13(16),14-clerodatetraen-18-oic acid, Hautriwaic lactone, Dodovisnoid G, Methyl-dodovisate B, 5α-Hydroxy-1,2-dehydro-5,10-dihydroprintziasaure-methylester, Strictic acid, Dodonolide [137]; Hautriwaic acid [138]; 2,18-Dihydroxylabda-7,13(*E*)-dien-15-oic acid, 5,7-Dihydroxy-3,6,4′-trimethoxy-3′-(4-hydroxy-3-methyl-but-2-enyl)flavone, 2,17-Dihydroxylabda-7,13(*E*)-dien-15-oic acid, 2-Hydroxylabda-7,13(*E*)-dien-15-oic acid, 3,6-Dimethoxy-5,7,4′-trihydroxyflavone, Penduletin, Santin [139].
*Eleocharis dulcis* (Burm.f.) Trin. ex Hensch. (Cyperaceae)	Whole plant infusion in saltwater (preferred for those growing in or near saltwater) is applied to wounds and sealed with a hollow stem of the same plant [25].	China	Whole plant; peel	6′-(4″-Hydroxy-3″-methoxy-phenylpropenyl)-1-(10-methoxy-phenylacetone)-1′-*O*-β-d-glucopyranoside, Susaroyside A, Clausenaglycoside A-D, Emarginone A and B, Thoreliin B, 4-*O*-(1′,3′-Dihydroxypropan-2′-yl)-dihydroconiferyl alcohol 9-*O*-β-d-glucopyranoside, 2-[4-(3-Methoxy-1-propenyl)-2-methoxy-phenoxy]-propane-1,3-diol, 6′-*O*-(*E*-Cinnamoyl)-coniferin, Methyl 3-(2-*O*-β-d-glucopyranosyl-3,4,5,6-tetramethoxyphenyl) propanoate, 9-*O*-(*E*-Cinnamoyl)-coniferin, 6′-*O*-(*E*-Cinnamoyl)-syringin, 2′-*O*-(*E*-Cinnamoyl)-syringin [140].
*Eucalyptus camaldulensis* Dehnh. (Myrtaceae)	Gum (or kino) mixed with water is taken orally (recommended not more than 1.3 g of kino) against diarrhoea; infusion made from aerial parts is used for washing head to heal colds and fevers [42,141,142].		NA	NA
*Euphorbia hirta* L. (Euphorbiaceae)	A decoction from dried herb (whole plant) is used for deworming, dysentery, bowel problems, and colic warts [31,42].	India	Whole plant	Kaempferol, Rutin, Quercetin [143].
*Euphorbia tirucalli* L. (Euphorbiaceae)	The plant is known for healing skin cancer [32].	China	Aerial; latex	12-*O*-(2*E*,4*E*,6*E*,8*E*-Tetradecatetraenoyl)-13-*O*-isobutyroyl-4β-deoxyphorbol, 13-*O*-acetyl-12-*O*-(2*Z*,4*E*-Octadienoyl)-4β-deoxyphorbol, Pedilstatin, 4β-Deoxy-phorbol-13-acetate, 4*α*-deoxy-phorbol-13-acetate, 3-*O*-(2,4,68-Tetradecatetraenoyl) ingenol [144].
*Excoecaria agallocha* L. (Euphorbiaceae)	Toxic juice from this plant is applied externally to relieve painful punctures caused by marine organisms, such as the sharp spines of some fish. Infusion from the bark is rubbed against body pain [32,42].	Australia, China, India, Japan, and Vietnam	Leaves; stems; resinous wood; roots; twigs; bark	12-Deoxyphorbol 13-(3*E*,5*E*-decadienoate) [145]; Excoecarins R1 and R2 [146]; 3α,11β-Dihydroxy-*ent*-isopimara-8(14),15-dien-2-one, 16β-Hydroxy-*ent*-atisan-3-one, Ribenone, *ent*-labda-8(17),13*E*-diene-3β,15-diol, *ent*-3β-Hydroxybeyer-15-*ene*-2,12-dione [147]; Excoecarins S, T1-T2, *ent*-12-oxo-2,3-Secobeyer-15-*ene*-2,3-dioic acid, *ent*-15-*epoxy*-Beyerane-3α-ol, Agallochin H [148]; Excoecarins V1—V3, 3,5,7,3′,5′-Pentahydroxy-2*R*,3*R*-flavanonol 3-*O*-α-l-rhamnopyranoside, *ent*-Atisane-16α-ol, *ent*-2,3-Secobeyer-15-ene-2,3-dioic acid, *ent*-15,18-Dihydroxybezoate, 3,4,5-Trimethoxyphenol 1-*O*-β-d-(6-galloyl)-glucopyranoside [149]; 3β-[(2*E*,4*E*)-5-oxo-Decadienoyloxy]-olean-12-ene, β-Amyrin acetate, Taraxerone, 3-Epitaraxerol, Epilupeol, Taraxerol, Taraxerone, 3β-[(2*E*,4*E*)-6-oxo-Decadienoyloxy]-olean-12-ene, Acetyl aleuritolic acid, Cycloart-22-*ene*-3β,25-diol, β-Sitostenone, (24*R*)-24-Ethylcholesta-4,22-dien-3-one, β-Sitosterol [150,151]; Excoagallochaols A–E [152]; Agallochins A-E [153,154]; Excoecarins D, E, and K [155]; Agallochins J-L [154,156]; Agallochins F-I, 2-Acetoxy-1,15-beyeradiene-3,12-dione, 2-Hydroxy-1,15-beyeradiene-3,12-dione, *ent*-kauran-16β-ol-3-one [148,154,157]; Excoecariphenols A-D [158]; Agallochaols K–P, Agallochaol Q, *ent*-17-Hydroxykaur-15-en-3-one, *ent*-Kaur-15-en-3β,17-diol, 7-Deoxogeayine, *ent*-15-Hydroxylabd-8(17),13*E*-dien-3-one, *ent*-15,18-Dihydroxylabd-8(17),13*E*-diene, *ent*-3β,11α-Dihydroxyisopimara-8(14),15-dien-2-one, *ent*-3β-Hydroxybeyer-15-en-2,12-dione [159]; *ent*-16α-Hydroxy-atisane-3,4-lactone, *ent*-16α-Hydroxy-atisane-3-one, *ent*-Atisane-3β,16α-diol, *ent*-3,4-seco-16α-Hydroxyatis-4(19)-en-3-oic acid [160]; Triacontane [161]; Agallochins M-P [159,162,163]; Excagallonoid A, *ent*-(3α,5β,8α,9β,10α,12α)-3-Hydroxyatis-16-en-14-one, Atis-16-ene-3,14-dione, 2-Hydroxy-*atis*-1,16-diene-3,14-dione, 12-Hydroxy-13-methylpodocarpa-8,11,13-trien-3-one [164]; Excolides A-B [165]; Afzelin, Quercitrin, Rutin, Kaempferol-3-*O*-(2-*O*-acetyl)-α-l-rhamnopyranoside, Kaempferide 3-*O*-α-l-rhamnopyranoside, Kaempferol 3-*O*-α-l-arabinofuranoside [166]; Agallolides A-M [167]
*Flueggea virosa* (Roxb. ex Willd.) Royle (Phyllanthaceae)	An aqueous leaf infusion is taken orally to heal internal pains, such as toothache; the liquid is applied to skin sores [42,168].	China and Taiwan	Aerial; roots	Flueggether A, Virosinine A [169]; Flueggenines A, B, and D, Norsecurinine [170,171,172]; Flueggines A and B [173]; Fluevirosines A-C [174]; Virosaines A and B [171,175]; 3β,12-Dihydroxy-13-methylpodocarpa-6,8,11,13-tetraene, 3β,12-Dihydroxy-13-methylpodocarpa-8,11,13-triene, Spruceanol, *ent*-3β,12α-Dihydroxypimara-8(14),15-diene, 3α-Hydroxy-12-methoxy-13-methyl-entpodocarp-6,8,11,13-tetraene, 3α-Hydroxy-13-hydroxymethyl-12-methoxy-ent-podocarp-6,8,11,13-tetraene, 3β-Hydroxy-13-hydroxymethyl-12-methoxy-ent-podocarp-6,8,11,13-tetraene, 12-Hydroxy-13-methylent-podocarp-6,8,11,13-tetraen-3-one, 12-Methoxy-13-methyl-ent-podocarp-6,8,11,13-tetraen-3-one, 6β,12-Dihydroxy-13-methyl-*ent*-podocarp-8,11,13-trien-3-one, 7α,20-Epoxy-3α-hydroxy-12-methoxy-13-methyl-*ent*-podocarp-8,11,13-triene, 3α,20-Epoxy-3β-hydroxy-12-methoxy-13-methyl-*ent*-podocarp-8,11,13-triene [176,177]; Fluvirosaones A and B, Virosecurinine [172,178]; 9(10→20)-Abeo-*ent*-podocarpane; 3,10-Dihydroxy-12-methoxy-13-methyl-9(10→20)-abeo-*ent*-podocarpa-6,8,11,13-tetraene; 4*E*-Dehydrochebulic acid trimethyl ester; 12-Hydroxy-20(10→5)-*abeo*-4,5-*seco*-podocarpa-5(10),6,8,11,13-pentaen-3-one; Betulinic acid 3β-calfeate, (+)-Ampelosin E [177]; Flueggrenes A and B [179]; Flueggenoids A–E, 6,12-Dihydroxy-13-methyl-7-oxo-*ent*-podocarpa−5,8,11,13-tetraeno-20,3α-lactone; 10α,12-Dihydroxy-13-methyl-9(10→20)-abeo-*ent*-podocarpa−6,8,11,13-tetraen-3-one; 12-Hydroxy-20(10→5)-abeo-4,5-seco-podocarpa-5(10),6,8,11,13-pentaen-3-one; Securinine, Bergenin, Norbergenin [171]; Fluevirines E and F, Viroallosecurinine [172]; Flueindolines A–C, Donaxanine, Methyltryptamine, *N*,*N*-Dimethyltryptamine, 1-Acetyl-β-carboline, 1-Hydroxymethyl-β-carboline, *N*-Methyl-1,2,3,4-tetrahydro-β-carboline, Strychnocarpine, Racemate, Hydromethyl-2-methyl-tetrahydro-β-carboline [180].
*Heliotropium ovalifolium* Forss (Heliotropiaceae)	Herb extract is used to relieve fevers [181].	India, Egypt, and Zimbabwe	Aerial	Heliophenanthrone [182]; Retronecine, Helifoline [183]; Supinine, 7-Angelyl-heliotridine [184]; 4,7,8-Trimethoxy-naphthalene-2-carboxylic acid, 6-Hydroxy-5,7-dimethoxy-naphthalene-2-carbaldehyde [185]; Heliotropamide [186].
*Hibiscus tiliaceus* L. (Malvaceae)	Infusions from bark and sapwood (with salt or freshwater) are applied to wounds and covered with the bark of the same plant [25,42].	China, Japan, and Taiwan	Stem; wood; bark	Hibiscusin, Hibiscusamide, Vanillic acid, 4-Hydroxybenzoic acid, Syringic acid, 4-Hydroxybenzaldehyde, Scopoletin, N-*trans*-Feruloyltyramine, N-*cis*-Feruloyltyramine [187]; 27-oic-3-oxo-28-Friedelanoic acid, 3α-Hydroxyfriedelane-2-one, 4α-Hydroxyfriedelane-3-one, Friedelin, Epifriedelanol, Pachysandiol A, 3β-*O*-(*p*-Hydroxy-*Z*-cinnamoyl)oleanolic acid, 3β-*O*-(*p*-hydroxy-*E*-cinnamoyl)oleanolic acid, oleanolic acid [188]; Hibiscusterpene I-V, Hibiscone B and C, Isohemigossypol-1-methyl ether, Virginicin, Parvifloral A, Syriacusin A [189].
*Ipomoea brasiliensis* (L.) Sweet (*I. pes-caprae* (L.) R. Br.) (Combretaceae)	Leaves decoction is applied externally for sores; the heated leaves are used to discharge boils [32,42].	China, India, Mexico, and Thailand	Whole plant	Pescapreins X-XVII [190]; β-Damascenone, Phytol [191]; Pescaproside A and B, Pescapreins I-IX, Stoloniferin III [192]; Ipomeolides A and B, Presqualene alcohol, Icosyl (*E*)-3-(4-hydroxyphenyl)acrylate, β-Sitosterol-3-*O*-β-d-glucopyranoside, Stigmasterol, Lupeol [193].
*Litsea glutinosa* (Lour.) C.B.Rob. (Heliotropiaceae)	Leaves and bark decoctions are applied to sores and to relieve body pain; sometimes, chewed leaves are applied to cuts and sores [32,34,42].	China and India	Leaves; twigs; heartwood	Glutin, β-sitosterol, Stigmasterol, (−)-Epicatechin, Sitosterol-β-d-glucopyranoside [194]; (3*R*,4*S*,5*S*)-2-Hexadecyl-3-hydroxy-4-methylbutanolide, Litsealactone C, D, and G, Eusmoside C [195].
*Macaranga tanarius* (L.) Müll.Arg. (Euphorbiaceae)	The plant is known for wound healing [196].	Japan, Taiwan, Thailand, and Vietnam	Bark; leaves; fruits; glandular trichomes	(2β,5β,10α,13α)-2-Hydroxypimara-9(11),15-*dien*-12-one, Methyl 2α-hydroxy-3β-[(4-hydroxybenzoyl)oxy]taraxer-14-en-28-oate, 2α-Acetoxy-3β-[(4-hydroxybenzoyl)oxy]-taraxer-14-en-28-oic acid, β-Sitosterol, Friedelin, Friedelin-3β-ol, β-Amyline, Macarangonol, 3β-Acetoxytaraxer-14-*en*-28-oic acid, 2α-Hydroxy-3β-[(4-hydroxybenzoyl)oxy]taraxer-14-*en*-28-oic acid [197]; (+)-Pinoresinol 4-*O*-[6″-*O*-galloyl]-β-d-glucopyranoside, Roseoside, Icariside B_5_, (6*R*,9*R*)-3-oxo-α-ionol β-d-glucoside, (6*R*,9S)-3-oxo-α-Ionol β-d-glucoside, (2*S*,3*R*)-Dihydrodehydrodiconiferyl alcohol β-d-glucoside, (+)-Pinoresinol 4-*O*-β-d-glucopyranoside, Scopoline, Rutin, Quercetin 3-*O*-galactopyranoside, Quercetin 3-*O*-arabinopyranoside, Isovitexin, Methyl gallate, Hexenyl β-d-glucoside, (*E*)-2-Hexenyl β-d-glucoside, Malloapeltine [198]; Macarangiosides A-F, Mallophenol B, Lauroside E [199]; Tanariflavanones A-D [198,200,201]; Macaflavanones A-G, Kolavenol [202]; 3′-Geranyl-naringenin [203]; Nymphaeol A-C, Isonymphaeol B, 3′-Geranyl naringenin [200,201,202,203,204]; Macatanarin D, Schweinfurthins E-H, and K-Q,5-((*E*)-3,5-Dihydroxystyryl)-3-((*E*)-3,7-dimethylocta-2,6-dien-1-yl)benzene-1,2-diol [205]; Tanarifuranonol, Vomifoliol, Blumenol B, vedelianin, mappain, methyl-mappain [201,206].
*Manihot esculenta* Crantz (Euphorbiaceae)	The plant is known to be effective against belly aches and diarrhoea [196].	NA	NA	NA
*Melaleuca leucadendra* (L.) L. (Myrtaceae)	The plant is known to be effective against headache, sinusitis, cough and colds, and skin sores [32,42].	Egypt	Essential oil	Stachyurin (or casuarinin), Ellagitannin [207].
*Merremia tridentata* (L.) Hallier f. (Combretaceae)	The whole plant is chewed or soaked in the water before applying it to the sores [131].	Vietnam	Stem bark	Apigenin, Cynaroside, Luteolin, Cosmosiin, Quercitrin [208].
*Morinda citrifolia* L. (Rubiaceae)	Leaves extract used to ease headaches [26,31].	French Polynesia and Japan	Fruits	(+)-3,4,3′,4′-Tetrahydroxy-9,7′α-epoxylignano-7α,9′-lactone, (+)-3,3′-Bisdemethyltanegool, (−)-Pinoresinol, (−)-3,3″-Bisdemethylpinoresinol, Quercetin, Kaempferol, Scopoletin, Isoscopoletin, Vanillin [209]; 1,5,15-Tri-*O*-methylmorindol, 2-*O*-(β-d-glucopyranosyl)-1-*O*-hexanoyl-β-d-gluropyranose, 2-*O*-(β-d-glucopyranosyl)-1-*O*-octanoyl-β-d-gluropyranose, 5,15-Di-*O*-methylmorindol, 1,3-Dihydroxy-2-methoxyanthracene-9,10-dione, 6-*O*-(β-d-Glucopyranosyl)-1-*O*-hexanoyl-β-d-glucopyranose, 6-*O*-(β-d-glucopyranosyl)-1-*O*-octanoyl-β-d-glucopyranose, 2,6-Di-*O*-(β-d-Glucopyranosyl)-1-*O*-hexanoyl-β-d-glucopyranose, 3-Methylbut-3-*enyl*-β-d-glucopyranose, 3-Methylbut-3-*enyl*-6-*O*-β-d-glucopyranosyl-β-d-glucopyranose, Asperulosidic acid, Rutin [210,211]; Nonioside A, (2*E*,4*E*,7*Z*)-deca-2,4,7-trienoate-2-*O*-β-d-glucopyranosyl-β-d-glucopyranoside, Tricetin [211].
*Nauclea orientalis* (L.) L. (Rubiaceae)	Aqueous bark infusion is used for sore belly; it is also applied externally to relieve rheumatic pains; the wood infusion is used for relieving fevers [32,43].	China, Japan, Laos, Papua New Guinea, Thailand, and Vietnam	Heartwood; bark; leaves; stems; roots;	Noreugenin, Naucleoside [212]; Angustine, 18,19-Dihydroangustine, 10-Hydroxyangustine, 3,14,18,19-Tetrahydroangustine, Parvine, Angustoline [213]; Nauclealines A and B, Naucleosides A and B, Strictosamide, Vincosamide, Pumiloside, Kelampayoside A, β-Sitosterol, Sitosteryl β-d-glucoside [214,215]; Naucleaorals A and B [216]; 10-Hydroxystrictosamide, 6′-*O*-Acetylstrictosamide [215]; α-Pinene, Loganetin, Loganin, Sweroside, Grandifloroside, Methyl 3,4-dihydroxybenzoate, 4-Hydroxycinnamic acid, 3-(2,4-Dihydroxylphenyl)propanoic acid, Methyl 3-(2,4-dihydroxylphenyl)propanoate, Skimmin, Adicardin, Aloe emodin, Pinoresinol [217]; Naucleaorine, Epimethoxynaucleaorine, Strictosidine lactam, 3,4,5-Trimethoxyphenol, 3α-Hydroxyurs-12-*en*-28-oic acid methyl ester, 3α,23-Dihydroxyurs-12-*en*-28-oic acid, 3α,19α,23-Trihydroxyurs-12-*en*-28-oic acid methyl ester, Oleanolic acid [218]; Nauclorienine, Antirhine, Iso-antirhine, Alangine, Naucline, Neonaucline, Angustidine, Subditine [219].
*Nelumbo nucifera* Gaertn. (Nelumbonaceae)	Milky juice from leaves is used against diarrhoea [31].	China, India, and Japan	Flowers; rhizome; leaves; seed embryo	2α,24-Diacetoxy-3β-hydroxyolean-12-*en*-28-oic acid, Hyptatic acid A, Maslinic acid, Botulin, Lupeol [220]; (*R*)-Coclaurine, (*S*)-norcoclaurine, Quercetin 3-*O*-β-d-glucuronide [221]; Neferine [222,223]; Liensinine, Isoliensinine [224]; Betulinic acid [225].
*Ochrosia elliptica* Labill. (Apocynaceae)	Bark is known to be good for dysentery [26].	China and Egypt	Stems and leaves	10-Methoxyconolidine, Apparicine, Vallesamine, Yunnanensine A, Angustilodine, Isositsirikine, (−)-Echitainine, Pseudo akuammigine [226]; Ursolic acid [227,228]; Ellipticine, elliptinine, methoxyellipticine, reserpiline (elliptine) [229].
*Ocimum tenuiflorum* L. (Heliotropiaceae)	The plant is used to relieve fevers [230].	NA	NA	NA
*Phyllanthus urinaria* L. (Phyllanthaceae)	The plant is used against colds [26,131].	China and Taiwan	Whole plant	Phyllanthin, Phyltetralin, Trimethyl-3,4-dehydrochebulate, Methylgallate, Rhamnocitrin, Methyl brevifolincarboxylate, β-Sitosterol-3-*O*-β-d-glucopyranoside, Quercitrin, Rutin [231]; Geraniin [232]; Corilagin, Ellagic acid [233].
*Phragmites australis* (Cav.) Trin. ex Steud. (Plantaginaceae)	The plant is used to treat sore throat [234,235].	China	Roots	*N*-*p*-Coumaroyl serotonin, *N*-*p*-Coumaroyl-trypamine, phranisines A-B [236].
*Sarcostemma viminale* (L.) R. Br (Apocynaceae)	The plant is indicated for skin sores and eye complaints [237].	NA	NA	NA
*Scaevola taccada* (Gaertn.) Roxb. (Euphorbiaceae)	Leaves decoction is applied externally to skin sores [8,32].	Thailand	Fruits	Scataccanol, *ent*-ammirin, Nodachenetin, Marmesin, Xanthyletin, Umbelliferone, 4-Formylsyringol, 6-Hydroxy-7-methyl-1-oxo-4-carbomethoxy octahydrocyclopenta[*c*]pyran, Loganetin, Matairesinol, 2-(4-Hydroxyphenyl) 3-(3,4-dihydroxyphenyl)-2-propenoate [238].
*Scoparia dulcis* L. (Plantaginaceae)	Leaves infusion is taken orally to heal stomach pain; the pulped whole plant is used for covering sores and cuts to enhance healing [32].	Bangladesh and Brazil	Whole plant	Glutinol [239]; Scoparinol [240]; *iso*-dulcinol, 4-*epi*-scopadulcic acid B, dulcidiol, scopanolal, dulcinol, and scopadiol [241].
*Terminalia catappa* L. (Combretaceae)	The plant is indicated for sore throat [196].	China and New Caledonia	Leaves; bark	Ursolic acid, 2,3,23-Trihydroxyurs-12-en-28-oic acid [242]; 3,4,5-Trimethoxyphenyl-1-*O*-(4-sulfo)-β-d-glucopyranoside, Chebuloside II, Arjunoglucoside II, Arjunolic acid, Betulinic acid, β-Sitosterol-3-*O*-β-d-glucopyranoside [243].
*Terminalia muelleri* Benth. (Combretaceae)	The plant is indicated for skin sores [196].	Egypt	Leaves	Apigenin-8-C-(2″-*O*-galloyl) glucoside 1, Luteolin-8-C-(2″-*O*-galloyl) glucoside 2, 1-*O*-Galloyl-2,3,4,6-dihexahydroxydiphenoyl-β-d-glucopyranoside, 1,4,6-Tri-*O*-galloyl-2,3-hexahydroxydiphenoyl-β-d-glucopyranoside, 1,2-Di-*O*-galloyl-4,6-hexahydroxydiphenoyl-β-d-glucopyranoside, Isostrictinin, 1-*O*-Galloyl-β-d-glucopyranoside, Combretum caffrum, Ellagic acid, Gallic acid [244,245]; Isoorientin, Vitexin, Chebulinic acid [245].
*Verbena officinalis* L. (Verbenaceae)	A decoction made from the whole plant is applied externally to overcome fever and rheumatic pain [31,42,246].	China and India	Aerial	3,4-Dihydroverbenalin, Daucosterol [247]; Ursolic acid [248]; Verbenalin, Hastatoside, Acteoside, β-sitosterol-d-glucoside [249].

Abbreviation: NA, not available.

## 4. Overview of the Anti-Inflammatory Mechanism of Action/Pathways

Bioactive crude extracts or isolated compounds from medicinal plants used by Aboriginal people attenuate inflammatory conditions through three major mechanisms, namely, (i) inhibition of nuclear factor kappa B (NF-κB) pathway, (ii) inhibition of iNOS (inducible nitric oxide synthase), and cyclooxygenase (COX-1 and COX-2) protein expression/activity, and (iii) inhibiting or downregulating NO production and various pro-inflammatory cytokines (such as interleukins, IL-1β, IL-6, interferon, IFN-γ, and tumour necrosis factor, TNF-α) or upregulating anti-inflammatory cytokines (such as IL-10 and transforming growth factor, TGF-β) (Figure 4).

Briefly, NF-κB is a key to inducing gene expression of many pro-inflammatory cytokines (including IL1b, IL-6, IL-12p40, TNF-α, and COX-2) in innate and adaptive immune cells [250]. NF-κB proteins consist of five members, including NF-κB1 (or p50), NF-κB2 (or p52), ReIA (or p65), ReIB, and c-ReI, and NF-κB activation occurs in two signalling pathways, canonical and alternative non-canonical pathways [251], where both pathways are involved in regulating immune and inflammatory responses. Innate immune cells (e.g., macrophages, dendritic cells, and neutrophils) are central to innate immunity and inflammation. Innate immune cells have PRRs (pattern recognition receptors) that can detect microbial products, including pathogen-associated molecular patterns (PAMPS) and also damage-associated molecular patterns (DAMPS)—released by damaged cells and tissues [252]. Five types of PRRs expressed by mammalian cells are toll-like receptors (TLRs), RIG-I-like receptors, NOD-like receptors (NLRs), C-type lectin-like receptors, and cytosolic DNA sensors [251], each with distinct structures to bind with different PAMPs and DAMPs. All PRRs share a similar signalling pathway by activating the canonical NF-κB pathway, and this pathway either directly induces inflammation by expressing pro-inflammatory cytokines and chemokines or indirectly promoting inflammatory T cells differentiation [253]. In response to different PAMPs and DAMPs, macrophages are activated and subsequently differentiate into M1 or M2, followed by the secretion of an array of cytokines and chemokines [250]. Pro-inflammatory cytokines such as IL-1, IL-6, IL-12, and TNF-α are characteristic of M1 macrophages. TLR signals play a vital role in regulating macrophage polarization, and for instance, TLR4 ligand lipopolysaccharide (LPS) promotes the formation of M1 phenotypes (Figure 4a) [250,254]. LPS is a bacterial-derived lipopolysaccharide that can induce inflammation in macrophage cells in in vitro assays. Activated M1 macrophages can also promote other inflammatory T cells, including Th1 and Th17 cells, and these cells, in turn, also mediate inflammation (Figure 4a).

Inhibiting NO production is another anti-inflammatory mechanism shown by many compounds isolated from selected medicinal plants. In mammalian cells, NO is mainly produced from the l-arginine:NO metabolic pathway by the enzyme called nitric oxide synthase (NOS), which has three isoforms of NOS—eNOS (endothelial NOS), nNOS (neuronal NOS), and iNOS [255]. eNOS and nNOS produce a controlled amount of NO in endothelial cells and neurons, respectively, under the Ca^2+^/calmodulin system [256] (Figure 4c), while iNOS produces NO only upon activation by specific cytokines (e.g., IFN-γ) or microbial products (e.g., LPS). Sustained NO production enhances the formation of reactive nitrogen oxide species (RNOs), and prolonged exposure to such free radicals is harmful to healthy body cells [257], whereby iNOS must be regulated.

**Figure 4 molecules-27-03849-f004:**
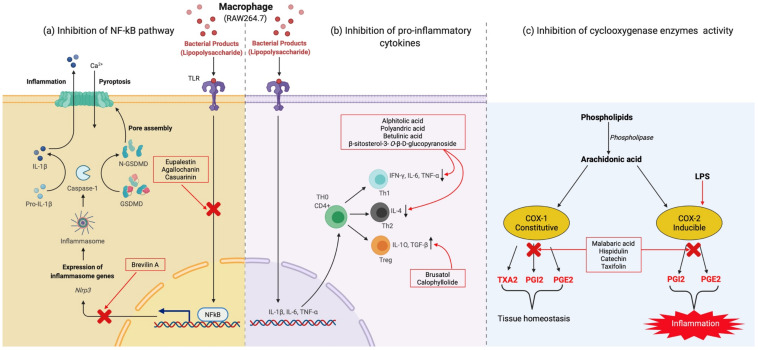
Three major mechanisms of anti-inflammatory activities shown by compounds isolated from medicinal plants of Tropical Australia: (**a**) inhibition of NF-κB pathway; (**b**) inhibition of various pro-inflammatory cytokines; (**c**) inhibition of cyclooxygenase enzymes (COX-1 and COX-2) activities. Abbreviations: COX (COX-1 and 2), cyclooxygenase 1 and 2; IL, interleukins; GSDMD, Gasdermin-D; LPS, lipopolysaccharide; NF-κB, nuclear factor kappa B; Nlrp3, NOD-, LRR- and pyrin domain-containing protein 3; PG, prostaglandins (PGI2, PGE2); Th cells, T-helper cells (Th1, 2); Th0, naïve T helper cells; Treg, regulatory T helper cells; TX, thromboxane (TXA2); TGF, transforming growth factor (TGF-β); TNF, tumour necrosis factor (TNF-α); TLR, toll-like receptor. *Courtesy*: Figure is adapted from Simon [258] and McDaniel et al. [259], and modified with web-based tool BioRender.

Cyclooxygenase enzymes break down arachidonic acid and form prostaglandin H_2_ (PGH_2_), subsequently catalysed into PGE_2_ and prostacyclin (PGI_2_) (Figure 4c) [260], and they mediate inflammations. COX-1 activity (constitutive) is responsible for maintaining vital physiological functions, while COX-2 activity (inducible) is usually activated by adequate stimuli coming from any damaged cells of inflammatory conditions [261].

## 5. Phytochemistry and Pharmacology of Medicinal Plants

### 5.1. Anti-Inflammatory Crude Extracts

Out of 45 anti-inflammatory medicinal plants included in this review, crude extracts from 30 species were already tested for anti-inflammatory activities in both in vitro and in vivo assays (Table 1). Pure compounds from 15 species have also been isolated and tested for their anti-inflammatory activities to validate their ethnopharmacological uses. Among 30 species, crude extracts of *Acalypha wilkesiana*, *Brucea javanica*, *Centipeda minima*, *Euphorbia hirta*, *Melaleuca leucadendra,* and *Terminalia catappa* are most widely studied against different inflammatory conditions (Table 2). Most of the studies on crude extracts have shown that they inhibit NO, PGE_2_, iNOS productions, and COX-2 expression in murine macrophage cells (e.g., RAW 264.7 cells) stimulated with bacterial LPS. Moreover, they also inhibit the production of pro-inflammatory cytokines, mainly TNF-α, IL-1β, and IL-6. 

For example, when LPS-induced RAW 264.7 cells were treated with an aqueous crude extract of *C. minima*, there was a significant decrease in NO production at a 100 μg/mL concentration and also reduced inflammatory cytokines levels (TNF-α and IL-1β) significantly [262]. Moreover, the aqueous extract also inhibited the expression of iNOS and COX-2 proteins by 80.2% and 71.2%, respectively, when incubated with LPS-activated RAW 264.7 cells for 24 h. The extract also significantly inhibits the expression of iNOS and COX-2 proteins in carrageenin-induced mice paw oedema [262]. Chan et al. [263] also observed a significant decrease in the expression of monocyte chemokine attractants, particularly CCL8, in LPS-stimulated RAW 264.7 cells by the crude extract of *C. minima*, which could have contributed to the inhibition of monocyte chemotaxis and macrophage infiltration in DSS (dextran sodium sulphate)-induced acute colitis in C57BL/6J mice. The crude extract also inhibits the LPS-induced production of TNF-α and IL-1β, protecting HT22 (immortalized mouse hippocampal cell line) neuronal cells from inflammatory damage [264]. Crude extracts from *A. wilkesiana*, *E. hirta*, and *M. leucadendra* also showed a similar anti-inflammatory activity (Table 2).

When Huang et al. (2017) studied the effect of oil emulsion from *B. javanica* in DSS (Dextran Sodium Sulphate)-induced acute colitis mouse model (0.5, 1, and 2 g/kg), oil emulsion improved disease activity index, including colon length, and body weight [265]. Additionally, when they analysed cytokines production, *B. javanica* oil emulsion at higher concentrations (1 and 2 g/kg) has significantly (*p* < 0.01) lowered the levels of six inflammatory cytokines (IL-1β, IL-6, IL-8, IL-17, IFN-γ, and TNF-α) in the colon tissues when compared to positive controls (sulfasalazine and azathioprine) [265]. Crude extracts from *T. catappa* bark, which Aboriginal people use to treat a sore throat, were anti-inflammatory. When Daram et al. (2021) compared ethanol and water extracts from *T. catappa* bark, ethanol extract was better in inhibiting 50% of protein denaturation in the in vitro egg-albumin denaturation assay, and diclofenac was used as a positive control [266]. Both extracts reduced carrageenan-induced paw oedema at a 500 mg/kg concentration in the rat. Leaves ethanol extract and chloroform fraction from *T. catappa* were tested in chronic and acute models of 12-*O*-tetradecanoylphorbol-13-acetate (TPA)-induced ear oedema. When crude extracts were applied topically at a 1 mg/kg concentration twice daily for four days, the chloroform fraction reduced oedema by 60% compared to ethanol extract (32%) [242]. In the same study, chloroform fraction yielded two pure compounds showing anti-inflammatory activities, discussed in the subsequent sections.

### 5.2. Anti-Inflammatory Compounds

When 45 selected Aboriginal medicinal plants were reviewed for their phytochemical compositions and pharmacological properties, 40 species were studied for their phytochemistry. For rest of the five species, only crude extracts were studied. When compounds isolated from 40 Aboriginal medicinal plants (Table 1) were further reviewed, 83 compounds have shown various anti-inflammatory activities (Table 2) in in vitro cellular and in vivo animal models. Out of 83 anti-inflammatory compounds, majority were terpenes and terpenoids (30 compounds), followed by flavonoids (16 compounds), coumarins (10 compounds), alkaloids (6 compounds), glycosides, sterols, lignans, and carboxylic acids (3 compounds each). The rest of the compounds were phenolics, aldehydes, tannins (2 compounds each), pyrans, phenylpropanoids, and fatty acid esters (one compound each).

#### 5.2.1. Terpenes and Terpenoids

Terpenes are one of the most structurally diverse secondary metabolites in plants, and five-carbon isopentyl units (2-methyl-1,3-butadiene or isoprene) form their main skeleton (additional functional groups such as ketone and hydroxyl rings are present in terpenoids), which are categorized into seven types based on a number of isoprene units they contain [14]. Among terpenoids, clerodane diterpenoid and polyandric acid A (isolated from Australian *Dodonaea polyandra*) (Figure 5) have shown potent anti-inflammatory activity [133]. Simpson et al. [133] tested this compound in acute and chronic mice models of TPA-induced ear oedema. Mice with TPA-induced ear oedema, when treated with 0.9 μM/ear of polyandric acid, significantly reduced (*p* < 0.05) the IL-1β (78%) compared to its negative control (TPA only treated group) [133], but it did not have any effect on TNF-α. In the chronic mouse ear oedema model, treatment of TPA-induced mice with polyandric acid A (same dosage, for four days) resulted in a significant reduction (*p* < 0.001) in the oedema (41.1%), where positive control drug betamethasone was reduced to 54.2%. In the same study, they further evaluated the inhibition of neutrophil infiltration into the inflamed tissues of TPA-induced mice ear oedema (i.e., assessed by measuring the level of MPO-myeloperoxidase), and polyandric acid A reduced MPO level by 81.5% (*p* < 0.001) compared to the negative control group. Positive control (betamethasone) showed 90% reduction of MPO at the same concentration (0.9 μM/ear) [133]. The compound also significantly reduced (*p* < 0.05) IL-β, proIL-β, and IL-6 by THP-1 monocytes and primary neonatal human epidermal keratinocytes (nHEKs) at both high (30 μM) and low (3 μM) concentrations. Hautriwaic acid, isolated from Mexican medical plant, *Dodonaea viscosa*, also reduced TPA-induced ear oedema in mice by 60.2%, 70.2%, and 87.1% at 0.25, 0.5, and 1.0 mg/ear concentrations [138]. When Feng et al. (2020) evaluated the ‘drug-likeliness’ of 65 anti-inflammatory clerodane diterpenoids (compared with marketed drugs) using SwissADME, a web-based tool by the Swiss Institute of Bioinformatics [267], many of them fulfill parameters within SwissADME [268]. They found that a lactone ring between C-18 and C-19 is mainly responsible for anti-inflammatory activity.

Another terpenoid, brusatol (Figure 5), isolated from aerial parts of Chinese grown *Brucea javanica*, showed anti-inflammatory activity by inhibiting the NF-κB pathway (Table 2). Incubation of brusatol (at three different dosages—25, 50, and 100 nM) with LPS-activated RAW 264.7 macrophage significantly inhibited the expression of NF-κB p65 protein [269]. Brusatol has effectively blocked the accumulation of p65 in the cells when analysed via Western blot; the inhibition was compared with the positive control indomethacin (10 μM) [269]. As described above, NF-κB p65 protein expression is responsible for initiating canonical NF-κB pathway. In parallel with the in vitro assay, brusatol has also significantly attenuated pathology in TNBS (trinitrobenzene sulfonic acid)-induced colitis in mice, including diarrhea, colon length, macroscopic damage, and histological injury. It also increased anti-inflammatory cytokines (TGF-β and IL-4), decreased the contents of IL-1β and IL-18, and elevated the levels of CAT, GSH and SOD in the colons [269]. From the roots of Thai *Brucea javanica,* seven quassinoids (brujavanol A and B, bruceine, 11-dehydroklaineanone, 15β-hydroxyklaineanone, 14,15β-dihydroxyklaineanone, 15β-*O*-acetyl-14hydroxyklaieanone [69].

Agallolides I and J from *Excoecaria agallocha* [154,167] showed a similar inhibitory effect on the NF-κB pathway (Table 2). Some isolated terpenoids also interrupted inflammations induced by activated RIG-I-like receptors and NOD-like receptors (NLRs) in the cytoplasm. Brevilin A (or 6-*O*-angeloylplenolin) isolated from *Centipeda minima* inhibits NLRP3 inflammasome in LPS-primed mouse bone marrow-derived macrophages via checking ROS formation [270]. The effect of brevillin A in inhibiting NLRP3 inflammasome was further evaluated in the LPS-induced mouse model. It significantly reduced the number of macrophages and neutrophils compared to MCC950 (small molecule NLRP3 pathway inhibitor) at a 20 mg/kg dosage [270]. Other compounds, including betulinic acid from *Nelumbo nucifera* [225], *Scoparia dulcis* [271], *Alphitonia excelsa* [51], and *Alphitonia netriei* [52], and alphitolic acid isolated from *A. excelsa* and *A. netriei* [52] (Table 2) inhibited or downregulated the secretin of various pro-inflammatory cytokines such as IL-1β, IL-4, IL-6, and TNF-α mostly in LPS -activated RAW 264.7 murine macrophage cells [225]. Anti-inflammatory activities of the rest of the terpenes and terpenoids are given in Table 2.

#### 5.2.2. Flavonoids

Flavonoids comprise seven different types, namely, anthocyanins, flavanols, flavanones, flavones, isoflavonoids, and flavan-3-ols (or catechins) [14]. Flavonoids are known to possess anti-inflammatory potential by inhibiting the expression of transcription factors and regulatory enzymes involved in the inflammation [272,273]. Sixteen flavonoids isolated from medicinal plants used by Aboriginal people of Australia have shown various anti-inflammatory activities (Figure 6; Table 2). However, most of these compounds were isolated from medicinal plants from other parts of the world, not from the Australian tropical region. Hispidulin isolated from the leaves of Thai *Clerodendrum inerme* inhibited NO production in LPS-stimulated RAW 264.7 cells with an IC_50_ value of 43.7 ± 4.0 μM [274]. Moreover, it also showed NF-κB pathway inhibitory effect. Agallochanin K from *Excoecaria agallocha* [154,167] showed a similar activity of NF-κB pathway inhibition (Table 2). Hispidulin was also reported from the Chinese *C. inerme* [105], but it was not tested for anti-inflammatory activity. There is no study on *C. inerme* from the tropical region of Australia.

The whole mashed plant of *Ageratum conyzoides* is used by Australian Aboriginal people for wound healing purposes [32,42]. Faqueti et al. (2016) isolated 5′-methoxy nobiletin and eupalestin from Brazilian *A. conyzoides* [275]. When administered before pleurisy in mice (induced by carrageenin), both compounds inhibited the phosphorylation level of NF-κB p65 and p38 MAPK in the lungs compared with both negative and dexamethasone-treated positive controls [276].

Flavonoids are also known to inhibit cyclooxygenase (COX-1 and COX-2) enzyme activity. Boeravinone N from *Boerhavia diffusa* [277], cianidanol (or catechin), taxifolin, aromadendrin, and ferrerol from *Corymbia terminalis* [132] are a few examples of flavonoids capable of inhibiting both COX-1 and COX-2 enzymatic activities (Table 2). Bairwa et al. (2013) tested boeravinon N (dosage, 40 *µ*M) isolated from Indian *B. diffusa* roots and obtained COX-1 and COX-2 inhibition at IC_50_ values 21.7 ± 0.5 and 25.5 ± 0.6 µM, respectively, when analysed using the COX (ovine) inhibitor screening kit [277], where indomethacin and celecoxib were used as positive controls. In the same study, another retinoid flavonoid, boeravinone B (dose, 50 mg/kg), was tested for inhibition of carrageenin-induced rat paw oedema, and it significantly inhibited 48.3 ± 1.7% (*p* < 0.001) of oedema at 3 h and 56.7 ± 1.6% (*p* < 0.01) at 5 h compared to positive controls celecoxib (35.8 ± 1.6% at 3 h and 43.5 ± 2.4% at 5 h) and ibuprofen (42.7 ± 2.7% at 3 h and 50.4 ± 2.2% at 5 h) [277].

#### 5.2.3. Alkaloids

Alkaloids contain nitrogen atom(s), and seven types of alkaloids (based upon their amino acid precursor) are present in at least 20% of plant species [14]. Plant-derived alkaloids are promising against inflammatory bowel diseases as many of them can restore epithelial barrier functions, ameliorate colonic inflammation, and reduce oxidative stress [278]. Plant families, such as Papaveraceae, Ranunculaceae, Solanaceae, and Amaryllidaceae, are rich in alkaloids. Among 45 selected reviewed here (Table 1), *Crinum pedunculatum* is the only species that belongs to alkaloid-rich families (Amaryllidaceae), and Aboriginal people use this whole plant against inflammation caused by marine stings. There are no reports of its phytochemistry yet, and this plant may contain novel anti-inflammatory alkaloids. When Barbosa-Filho et al. [279] reviewed anti-inflammatory properties of 171 alkaloids, 137 of them were found to possess anti-inflammatory properties, of which isoquinoline alkaloids were reported to be mostly bioactive. A total of six alkaloids isolated from Chinese medicinal plants *Ochrosia elliptica* (ellipticine, 10-methoxyconolidine, apparicine, yunnanensine, and isositsirikine) and *Nelumbo nucifera* (neferine) were reported as anti-inflammatory (Figure 7; Table 2). Indole alkaloid ellipticine at a concentration of 5 μM downregulated the TNF-α and IL-6 levels in LPS-stimulated RAW 264.7 cells and human peripheral blood monocytes [280]. When Chen et al. (2017) assessed four alkaloids isolated from leaves of *O. elliptica*, 10-methoxyconolidine and apparicine showed significant inhibition of nitric oxide production in LPS-stimulated RAW 264.7 cells at IC_50_ values of 2.4 ± 0.08 μM and 5.8 ± 0.13 μM, respectively. However, the other two alkaloids (yunnanensine and isositsirikine) showed only mild inhibition [226]. Aboriginal people of Australia have used the bark of this plant for treating dysentery, but there is no report of compounds isolated from the bark. Neferine, a benzylisoquinoline alkaloid from the seed embryo of Chinese *Nelumbo nucifera*, also showed a similar anti-inflammatory activity by inhibiting TNF-α and IL-6 in LPS-induced RAW 264.7 cells; additionally, it suppressed COX-2 and iNOS expression, and DSS-induced ulcerative colitis in the mice (at dosages of 10 and 25 mg/kg) [281]. Min et al. (2019) showed that neferine protected mice against DSS-induced colitis at concentrations of 5 and 10 mg/kg [282]. Neferine further reduced MPO activity and secretion of inflammatory cytokines (TNF-α, IL-6, and IL-1β).

There are many reported cases of poisoning from the consumption of medicinal plants and herbal products, and the presence of alkaloids such as pyrrolizidine alkaloids (PAs) is responsible for the toxicity [283,284]. Plants belonging to Asteraceae, Fabaceae, and Boraginaceae are known to contain PAs. When Wang et al. (2021) analysed 386 herbal products from the Chinese Pharmacopoeia (2020) for the presence of PAs using ultra-high performance liquid chromatography-tandem mass spectrometry (UPLC-MS/MS), 271 of their tested samples contained 0.1–25,567.4 μg/kg, out of which 20 samples had PAs above the estimated daily intake value of 0.007 μg/kg body weight/day [285]. Senkirkine (in 189 out of 271 samples), intermedine (in 122 out of 271 samples), and lycopsamine-*N*-oxide were the top three PAs detected in their test samples. A similar study was also conducted by Letsyo et al. (2017) on 70 popular medicinal plant-derived products widely sold in Ghana and West African countries, and 60% of the tested samples contained PAs with average concentration of 25 μg/kg [286]. One must be aware of their detoxification processes before utilizing them in any form of medicinal preparations because, as per the World Health Organisation (WHO), daily intake of PAs above 10 μg/kg body weight can cause hepatic veno-occlusive disease [287]. A C19 norditerpenoid alkaloid aconitine from *Aconitum* root is another example of a toxic alkaloid and the *Aconitum* species are widely used in traditional medicine in Asian countries [284]. Only two plants (*Ageratum conyzoides* and *Centipeda minima*) out of 45 species reviewed here belonged to PAs containing family (Asteraceae). *A. conyzoides* contains PAs lycopsamine, dihydrolycopsamine, and acetyl-lycopsamine [288]. The toxic effect of hydroalcoholic extract of *A. conyzoides* leaves in Caco-2 cells (IC_50_ = 44 μg/mL) is attributed to pyrrolizidine alkaloid content [289]. However, both plants are indicated for external applications for wounds and eye inflammations by Aboriginal people, which may be considered safe, but it would not be advisable for oral consumption unless processed properly, as PAs are hepatoxic and carcinogenic. Aboriginal healers have extensive knowledge of detoxifying some toxic plants before eating or using them in medical applications [46,47], but it requires adequate documentation.

**Table 2 molecules-27-03849-t002:** List of medicinal plants and their crude extracts and/isolated compounds showing various anti-inflammatory activities.

Plant Species	Crude Extracts Tested	Isolated Compounds Tested	Anti-Inflammatory Activities
*Acalypha wilkesiana*	Leaves extract	NT	Suppressed lipopolysaccharide (LPS)-induced nitric oxide (NO), prostaglandins E2 (PGE_2_), and inducible nitric oxide synthase (iNOS) productions, and cyclooxygenase-2 (COX-2) expression in RAW 264.7 cells; also reduced secretion of tumour necrosis factor-alpha (TNF-α), interleukins 1beta (IL-1β), and IL-6 in LPS-stimulated RAW 264.7 cells [290]; also attenuated carrageenin-induced inflammation/oedema [291,292].
*Ageratum conyzoides*	Leaves extract; aerial extract	Coumarin, 5′-Methoxy nobiletin, and Eupalestin	Leaves extract is anti-inflammatory in subacute (cotton pellet-induced granuloma) and chronic (formaldehyde-induced arthritis) models of inflammation in rats [293]; also reduced paw oedema [275]; pure compounds reduced p65 nuclear factor kappa B (NF-κB) and p-p38 mitogen-activated protein kinase (MAPK) [276] activities.
*Alphitonia excelsa*	NT	Betulinic acid	Reduced levels of COX-2, NO, TNF-α, and IL1-β in tissues obtained from λ-carrageenan-induced paw oedema mice [294].
*Alphitonia petriei*	NT	Alphitolic acid, *trans*- and *cis*-Coumaroyl esters of alphitolic acid, and Betulinic acid	Reduced or inhibited NO production and TNF-α level in LPS + interferon-gamma (IFN-γ) activated RAW 264.7 cells [52].
*Angophora costata*	NT	Costatamins A-C	Reduced NO production and TNF-α secretion in RAW 264.7 cells [54].
*Antidesma bunius*	NT	Antidesoside, Podocarpusflavone A, and Amentoflavone	Reduced NO production in LPS-stimulated BV2 cells and RAW 264.7 cells [55].
*Barringtonia racemosa*	Inflorescence axes extract	Barringoside I	Inflorescence axes extract inhibited xanthine oxidase (XO) activity [295]; fruit extract—attenuated acute inflammation induced by inflammogens in rat paw oedema and also in carrageenin-induced rat paw oedema [296]. Barringoside I moderately inhibited LPS-induced NO production in RAW 264.7 cells [65].
*Brasenia schreberi*	NT	Quercimeritrin	Reduced the expression of iNOS and NO in LPS-stimulated RAW 264.7 cells; also prevented the overexpression of COX-2 and granulocyte macrophage-colony-stimulating factor [66].
*Brucea javanica*	Oil emulsion; seeds extract	Brusatol	Oil emulsion attenuated pathology in dextran sodium sulphate (DSS)-induced colitis in mice and reduced levels of TNF-α, IL-1β, IL-6, IL-8, IL-17, and IFN-γ [265]; seed extract inhibited the production of NO, PGE2, TNF-α, IL-1β, and IL-6 but increased anti-inflammatory IL-10 cytokine [297]. Brusatol reduced TNF-α, pro-IL-1β, PGE_2_, and NO levels; also suppressed NF-κB signalling pathway in LPS-stimulated macrophages; significantly attenuated pathology in 2,4,6-trinitrobenze sulfonic acid (TNBS)-induced mice colitis; suppressed IL-1β and IL-18 levels, and elevates levels of catalase (CAT), glutathione (GSH), and superoxide dismutase (SOD) enzymes in the colon tissue [269].
*Calophyllum inophyllum*	Leaves extract	Calophyllolide, and 27-[(*E*)-*p*-coumaroyl] canophyllic acid	Leaves extract suppressed LPS-induced NO production, and the expression of iNOS, COX-2, and NF-κB in RAW 264.7 cells [298]. Pure compounds downregulated IL-1β, IL-6, TNF-α, and NO production but upregulated IL-10 in RAW 264.7 cells [82,87].
*Centella asiatica*	Whole plant extract	Asiatic acid, Isomadecassoside, Asiaticoside G, 11-oxo-Asiaticoside B, and Rosmarinic acid	Crude extract reduced IL-13 and inhibited activation of NF-κB pathway [299,300]. Pure compounds reduced NO production in LPS-stimulated RAW 264.7 cells [94,98,299].
*Centipeda minima*	Whole plant extract	Brevilin A, Centiplide A, Centiplides H, and Helenalin-isovalerate	Whole plant extract reduced NO production in LPS-induced RAW 264.7 cells and λ-carrageenan-induced paw oedema [262]; also inhibited monocyte chemotaxis and macrophage infiltration in DSS-induced acute colitis mouse model [263]; also inhibited the LPS-induced production of TNF-α and IL-1β [264]. Pure compounds attenuated LPS-induced NF-κB pathway activation and oxidative stress, and thus, suppressed neuroinflammation [301]; also reduced NO production in LPS-activated RAW 264.7 cells [101]; reduced IL-1β secretion to suppress NOD-, LRR- and pyrin domain-containing protein 3 (NLRP3) inflammasome in LPS-induced macrophage cells and monosodium urate (MSU)-challenged peritonitis model [270].
*Cleome viscosa*	NT	Quercetin 3-*O*-(2″-acetyl)-glucoside, Cleomiscosins A-C, and Malabaric acid	Reduced carrageenan-induced rat paw oedema [113]; reduced IL-4, TNF-α, and NO production in LPS-stimulated mouse solenocytes [302]; also reduced COX-1 and 2 activities [114].
*Clerodendrum inerme*	Leaves extract	Hispidulin	Leaves extract inhibited NO production in LPS-stimulated RAW 264.7 cells [274]. Hispidulin inhibited PGE2 production, and iNOS and COX-2 expressions via the blockade of NF-κB DNA-binding activity and c-Jun N-terminal Kinase (JNK) pathway [274].
*Corymbia terminalis*	NT	Taxifolin, Aromadendrin, Cianidanol, and Farrerol	Suppressed IL-6 level in LPS-stimulated cells; also suppress IL-8 and COX-1 and 2 enzyme activities in keratinocytes [132].
*Crinum pedunculatum*	Bulb extract	NT	Bulb extract inhibited carrageenin-induced rat paw oedema [303].
*Dodonaea polyandra*	Leaves extract	Polyandric acid A, 15,16-Epoxy-8α-(benzoyloxy) methyl-2α-hydroxycleroda-3,13(16), and 15,16-Epoxy-2α-benzoyloxycleroda-3,13(16),14-trien-18-oic acid.	Leaves extract reduced 12-*O*-Tetradecanoylphorbol acetate (TPA)-induced mouse ear oedema [304]. Pure compounds’ topical application significantly reduced IL-1β production in mouse ear tissue in an acute model [133]; attenuated TPA-induced mouse ear oedema [135].
*Dodonaea viscosa*	Leaves extract	Hautriwaic acid.	Leaves extract reduced carrageenin-induced rat paw oedema [305]. Hautriwaic acid reduced inflammation in TPA-induced mice ear oedema [138].
*Eleocharis dulcis*	NT	Susaroyside A.	Reduced TNF-α level in LPS-activated macrophage cells [306].
*Eucalyptus camaldulensis*	Crude EO	NS	Reduced carrageenan-induced paw oedema, xylene-induced ear oedema, and cotton pellet-induced granuloma [307].
*Euphorbia hirta*	Leaves extract, whole plant extract	β-Amyrin.	Leaves extract inhibited TNF-α production in LPS-treated rats [308]; inhibited LPS-induced NO production in peritoneal macrophages [308]; inhibited NO production and iNO protein expressions in LPS-activated RAW 264.7 cells [309]; reduced carrageenin-induced oedema in rats [310]; inhibited PGE_2_ production in rabbit synovial fibroblast cells (HIG-82) [311]; whole plant extract inhibited NO production in LPS-induced RAW 264.7 cells [312]; reduced pro-inflammatory cytokines in adjuvant-induced arthritis mice [313]. β-amyrin inhibited the cellular molecules (E-selectin, sICAM-1, and sVCAM-1) involved in the development of atherosclerotic initiation induced by pro-inflammatory cytokines in SVEC4-10 endothelial cells via activation of the endothelial nitric oxide synthase (eNOS) and attenuation of adhesion molecules expressions [314].
*Euphorbia tirucalli*	Roots extract	NT	Roots extract reduced acetic acid-induced pain/inflammation by inhibiting TNF-α and IFN-γ productions [315].
*Excoecaria agallocha*	NT	Agallochaols K, O, and Q, *ent*-17-Hydroxykaur-15-*en*-3-one, *ent*-Kaur-15-*en*-3β,17-diol, *ent*-15,18-Dihydroxylabd-8(17),13*E*-diene, Agallolides I and J, and Agallochanin K.	Suppressed the expression of NF-κB and activator protein-1 (AP-1) targeted genes and TNF-α and IL-6 LPS-activated Raw 264.7 cells [154,159,167].
*Flueggea virosa*	NT	Flueggrenes A.	Inhibited superoxide anion generation and elastase release in N-Formylmethionyl-leucyl-phenylalanine (FMLP)/cytochalasin B (CB) activated-human neutrophils [179].
*Heliotropium ovalifolium*	NT	4,7,8-Trimethoxy-naphthalene-2-carboxylic acid and 6-Hydroxy-5,7-dimethoxy-naphthalene-2-carbaldehyde.	Reduced IL-6 and TNF-α in LPS activated human leukemia monocytic cell line (THP-1) cells [185].
*Hibiscus tiliaceus*	Whole plant extract	NT	Whole plant extract reduced acute auricle swelling induced by dimethylbenzene in mice [316].
*Ipomoea pes-caprae*	Stem and leaves extract	Eugenol and 2-Methoxy-4-vinylphenol.	Stems and leaves extract reduced trypsin-, histamine-, and bradykinin-induced paw oedema in mice [317]; inhibited prostaglandins synthesis [191].
*Litsea glutinosa*	Leaves extract	NT	Leaves extract reduced carrageenin-induced paw oedema in mice [318].
*Macaranga tanarius*	NT	Nymphaeol B.	Nymphaeol B inhibited COX-2 activity and reduced PGE_2_ production [201]
*Manihot esculenta*	Leaves extract	NS	Leaves extract reduced carrageenan-induced rat paw oedema and xylene-induced ear swelling in mice [319].
*Melaleuca leucadendra*	Whole plant extract	Stachyurin.	Whole plant extract suppressed LPS-induced NO and PGE2 production, and COX-2 expression in RAW 264.7 cells [320]. Stachyurin (or Casuarinin) reduced ethanol-induced gastric ulceration in rats by increasing mucin production and reducing acidity; it also increased glutathione and catalase levels; it suppressed the immunoexpressing of NF-κB, COX-2, and iNOS to their normal values [207].
*Merremia tridentata*	NT	Apigenin and Quercetrin.	Apigenin suppressed TNF-α, IL-1β, and IL-6 production in LPS-induced murine BV2 microglia cells, and also suppressed LPS-induced NF-κB pathway activation [321]; Quercetrin inhibited NO production and suppressed pro-inflammatory cytokines (TNF-α, IL-1β and IL-6) in LPS-induced RAW 264.7 cells [322].
*Morinda citrifolia*	Fruits extract; fruits juice; seeds extract	(+)-3,4,3′,4′-Tetrahydroxy-9,7′α-epoxylignano-7α,9′-lactone, (+)-3,3′-Bisdemethyltanegool, (−)-Pinoresinol, (−)-3,3′-Bisdemethylpinoresinol, Kaempferol, Scopoletin, Isoscopoletin, Vanillin, Asperulosidic acid, Rutin, Nonioside A, (2*E*,4*E*,7*Z*)-deca-2,4,7-Trienoate-2-*O*-β-d-glucopyranosyl-β-d-glucopyranoside, and Tricetin.	Fruits extract inhibited matrix metalloproteinase-9 (MMP-9) release from LPS-stimulated human monocytes [209]; fruit juice reduced both MPO activity and pro-inflammatory cytokines (TNF-*α* and IFN-*γ*) in the intestine of C57BL/6 mice exposed to DSS chemical [323]; fruits extract also reduced intracellular reactive oxygen species (ROS) and suppressed COX-2, IL-8, and PGE2 in Caco-2 cells, and neutrophil chemotaxis by suppressing the translocation of the p65 subunit [324]; seed extract inhibited NO production in LPS-stimulated RAW 264 cells [325]. Pure compounds decreased NO production, the expression of IKKα/β, I-κBα, and NF-κB p65 in LPS-stimulated macrophages [209,211].
*Nauclea orientalis*	Bark extract	NT	Bark extract protected the myocardium inflammation as a result of doxorubicin (Dox)-induced oxidative stress in Wistar rats [326].
*Nelumbo nucifera*	Flower extract; fruits extract; rhizome extract; leaves extract	Neferine, Quercetin, Cianidanol, and Betulinic acid.	Flowers extract suppressed TNF-α secretion in LPS-stimulated macrophages [327]; fruits extract—reduced carrageenin-induced paw oedema in rats [328]; also upregulated anti-inflammatory cytokines (IL-10 and 12), and downregulated pro-inflammatory cytokines (IL-6, IL-1β, TNF-α, and IFN-γ) [329]; rhizome extract reduced carrageenin- and serotonin-induced paw oedema in male albino Wistar rats [225]; leaves extract reduces the expression and productions of IL-1β, IL-6, TNF-α, PGE_2_, and NO; also reduced NF-κB activity by inhibiting NF-κB phosphorylation [330]. Pure compounds reduced LPS and LPS + a pan-caspase inhibitor (Z-VAD)-induced secretion of nitrite, inflammatory cytokines, and expression of iNOS and COX-2; oral administration of Neferine reduced inflammation in DSS-induced colitis model [281,282]; it also reduced IL-6 and TNF-α production in LPS-activated RAW 264.7 cells, and activated peroxisome proliferator-activated receptor (PPARα and PPARγ) [331]; quercetin 3-*O*-β-d-glucuronid reduced NO release in LPS-treated RAW 264.7 cells [332]; quercetin and Cianidanol inhibited JNK- and NF-κB-regulated pathways [333]; Betulinic acid reduced carrageenin and serotonin-induced rat paw oedema [225].
*Ochrosia elliptica*	NT	10-Methoxyconolidine, Ellipticine, apparicine, yunnanensine, isositsirikine	All compounds reduced the production of NO and pro-inflammatory cytokines (TNF-α and IL-6) in LPS-stimulated RAW 264.7 cells and human peripheral blood monocytes [226,280].
*Ocimum tenuiflorum*	Leaves extract	NS	Leaves extract reduced the LPS-induced inflammation in RAW 264.7 cells [334].
*Phragmites australis*	Aerial extract	NT	Aerial crude extract inhibited the production of NO and ROS, and pro-inflammatory cytokines (TNF-α and IL-1β) in LPS-induced RAW 264.7 cells [335].
*Phyllanthus urinaria*	NT	β-Sitosterol-3-*O*-β-d-glucopyranoside and Corilagin.	Both compounds inhibited the NO production in LPS- and IFN-γ-activated peritoneal macrophages [231]; decreased NF-κB/DNA interactions affecting the IL-8 gene expression in TNF-α treated IB3-1 cells. Inhibited TNF-α induced secretion of monocyte chemoattractant protein-1 (MCP-1) and RANTSE (Regulated on Activation, Normal T cell Expressed and Secreted) as well, suggesting its potential as an anti-inflammatory therapy for cystic fibrosis [336].
*Sarcostemma viminale*	Whole plant extract	NS	Whole plant extract inhibited the production of pro-inflammatory cytokines in RAW 264.7 cells [337].
*Scaevola taccada*	Leaves extract	NT	Leaves extract reduced IL-1β levels in the tissues of Sprague-Dawley rats suffering from mastitis [338].
*Scoparia dulcis*	Whole plant extract	Betulinic acid.	Whole plant extract reduced levels of COX-2, NO, TNF-α, and IL-1β in mice tissues from λ-carrageenan-induced paw oedema [271]; also inhibited both LOX and XO activities [294]. Betulinic acid reduced levels of COX-2, NO, TNF-α, and IL-1β in tissues obtained from mice with λ-carrageenan-induced paw oedema [294].
*Terminalia catappa*	Leaves extract, bark extract, stem extract	Ursolic acid and Asiatic acid (2α,3β,23-trihydroxyurs-12-en-28-oic acid).	Leaves extract reduced TPA-induced ear oedema in both acute and chronic male ICR (Institute of Cancer Research) mice models by inhibiting myeloperoxidase (MPO) activity [242]; bark extract reduced carrageenan-induced paw oedema by inhibiting cellular infiltration and MPO activity [266]; stem bark extract inhibited IL-1β and nitrite production in RAW 264.7 cells; also reduced colonic damage and weight/length ratio in TNBS-induced colitis model [339]. Ursolic acid and 2α,3β,23-trihydroxyurs-12-en-28-oic acid reduced TPA-induced ear oedema and inhibited MPO activity [242].
*Terminalia muelleri*	Leaves extract	NT	Leaves extract reduced carrageenan-induced paw oedema and lowers PGE_2_ and inflammatory cytokines (TNF-α, IL-1β, and IL-6) [245].
*Verbena officinalis*	Leaves extract	NT	Leaves extract reduced carrageenin-induced paw oedema in mice [249]; also reduced TPA-induced ear oedema [340].

Abbreviations: NS: compounds not isolated from the plant; NT: crude extracts not tested, or compounds isolated but not tested for anti-inflammatory activities.

#### 5.2.4. Coumarins

Costatamins A-C isolated from the leaves of Australian *Angophora costata* inhibited the production of NO and pro-inflammatory cytokine—TNF-α in LPS-activated RAW 264.7 cells with IC_50_ values in the range of 20–30 μg/mL [54]. Aboriginal people used an aqueous solution of reddish exudate from the bark/or trunk orally to cure diarrhoea [8,53], not leaves; thus, studying bark’s phytochemical constituents could help further validate their use. Calophyllolide (Figure 8) is one of the promising examples of coumarins from one of the selected medicinal plants. The compound isolated from fresh seeds of Vietnamese *Calophyllum inophyllum* showed wound-healing potential [82]. Aboriginal people have used grounded nut kernel to rub against body pain, but in Vietnam, *C. inophyllum* oil is used for treating skin and rheumatic disease and burns. Calophyllolide (6 mg/animal) accelerated the wound healing process in the surgically created wound in the mice model more than iodopovidone (100 mg/animal) at 14 days post-treatment, and it was determined that the reduced MPO and increased M2-related gene expression could have promoted the wound healing in mice [82]. Calophyllolide also downregulated pro-inflammatory cytokine IL-1β, IL-6, and TNF-α, and upregulated anti-inflammatory cytokine IL-10 in LPS-activated RAW 264.7 cells [82]. Based on these promising activities, calophyllolide could be a potent lead for developing topical wound-healing agents or creams. Other parts of the *C. inophyllum*, particularly those used for ethnomedicinal purposes, may also contain promising anti-inflammatory compounds. Scopoletin and isoceopoletin from fruits of *Morinda citrifolia* showed mild activity against 5-LO and 15-LO (lipoxygenase) and COX-2 enzymes, but IC_50_ values were low [209]. Other coumarins, cleomiscosins A-C from dried seeds of Indian *Cleome viscosa*, also showed anti-inflammatory activities by inhibiting pro-inflammatory cytokines (TNF-α and IL-6) when given in a mixture at concentrations of 30–100 mg/kg body weight in female Swiss albino mice [302] (Table 2).

#### 5.2.5. Glycosides 

Studies have reported the anti-inflammatory potential of glycosides [341], such as cardiac glycosides [342] and iridoid glycosides [343]. A few glycosides isolated from the medicinal plants have shown various anti-inflammatory activities. Quercimeritrin (or Quercetin-7-*O*-β-d-glucopyranoside) isolated from Canadian grown *B. schreberi* (Figure 9) was tested for cytotoxicity and anti-inflammatory activities. It was non-toxic to human skin fibroblast cells (WS-1) at 36 μg/mL with resulting 95% live cells [66]. For anti-inflammatory activity, quercimeritrin inhibited the expression of iNOS and COX-2 in LPS-stimulated RAW 264.7 cells by 90% and 100%, respectively, at 30 µg/mL concentrations [66]. However, the compound did not inhibit the TNF-α cytokine. Other two glycosides, (2*E*,4*E*,7*Z*)-deca-2,4,7-trienoate-2-*O*-β-d-glucopyranosyl-β-d-glucopyranoside from *Morinda citrifolia* and quercetin 3-*O*-(2″-acetyl)-glucoside from *Cleome viscosa* showed similar activity by inhibiting the secretion of pro-inflammatory cytokines and expression of inflammation mediators and proteins in the LPS-activated macrophages cells [209,211]. (2*E*,4*E*,7*Z*)-deca-2,4,7-trienoate-2-*O*-β-d-glucopyranosyl-β-d-glucopyranoside and tricetin (at a concentration of 50 μM), which were isolated from the fruit juice of *Morinda citrifolia* showed a significant effect on the LPS-induced expression of IKKα/β, IkBα, and NF-κB p65 in mouse macrophage RAW 264.7 cells [211]. IKKα and IKKβ are two catalytic subunits of I_K_B kinase (IKK), and this kinase initiates phosphorylation of I_K_Bα, which is linked to NF-κB pathway activation [251]. Co-incubation of (2E,4E,7Z)-deca-2,4,7-trienoate-2-*O*-β-d-glucopyranosyl-β-d-glucopyranoside and tricetin with LPS-activated RAW 264.7 cells significantly inhibited the expression of IKKα/β, IkBα, and NF-κB p65 when analysed by Western blot. Quercetin 3-*O*-(2″-acetyl)-glucoside reduced carrageenan-induced rat paw oedema [113].

## 6. Biodiscovery Potential and Challenges

Drug discovery from natural sources is a challenging and resource-intensive process. It requires high throughput screening (HTS) of large extracts, followed by tedious characterization and identification of bioactive constituents. Among different natural products, medicinal plants are still one of the vital sources of promising therapeutic leads. When Fabricant and Farnsworth (2001) evaluated 122 compounds used worldwide as drugs, 80% were associated with medicinal plants used in Indigenous medicines [344]. For example, taxol and camptothecin [345]. However, there is no evidence of income amount generated by commercialising these drugs to the Indigenous peoples who own intellectual property rights. Intellectual piracy/biopiracy incidences were rampant in the olden days, whereby the researchers/pharmaceutical companies stole the traditional knowledge without any financial returns to the indigenous communities. It is fair to respect the indigenous rights and equally share the benefits arising from value–addition to their knowledge. There is an urgent need to document Aboriginal people’s indigenous knowledge about medicinal plants and establish a proper database such that knowledge–ownership is secured in the hands of their younger generations.

Today, this has changed for a good cause. About two-thirds of medicinal plant species used worldwide (sourced from their natural habitats) [346] are strictly regulated by the equal benefit-sharing agreement that is highlighted in the Convention on Biological Diversity [347] and Nagoya Protocol [348], as well as local state and federal biodiversity protection acts. The plants associated with indigenous knowledge are bound by more stringent regulations, including the United Nations Declaration on the Rights of Indigenous People {Article 31(1)} [349], and local in-country biodiscovery acts and rules. All traditional knowledge-related developments, including cultural expression and biodiscovery, remain within their jurisdiction [349]. Moreover, indigenous intellectual property is their heritage, and they have the authority to approve or reject the consent to access as they must maintain secrecy [350]. In Australia, the Biodiscovery Act of Queensland 2004 [351] protects the IP and guides the biodiscovery involving Aboriginal medicinal plants, although only recognized in Queensland State.

This review has compiled 45 medicinal plants that Aboriginal people of Australia use to treat various inflammatory conditions. Some of these plant extracts and compounds showed potent anti-inflammatory activities, presenting unprecedented biodiscovery and drug development potential. In addition, most of the studied medicinal plants for their phytochemical and anti-inflammatory activities to date were reported from other countries, the majority from China, India, Japan, Taiwan, Thailand, and Vietnam. Australia reported only five species (*Alphitonia petriei*, *Angophora costata*, *Corymbia terminalis*, *Dodonaea polyandra*, and *Excoecaria agallocha*). Variations in the geography, habitats, and climate [14] can cause variations in plants’ chemical constituents and their associated bioactivities (antioxidant and anti-inflammatory) [352,353,354]. Thus, it is worthwhile to study the Australian medicinal plants, particularly those unstudied endemic medicinal plants, in collaboration with the traditional owners.

## 7. Conclusions

Australia is home to unique biodiversity, including many native and endemic species. The Aboriginal people of Australia, who are said to have lived from 40,000 to 50,000 years ago, possess profound oral traditional knowledge about their native flora and fauna. More than 900 medicinal plants are recorded in the tropical region alone, and associated traditional medicinal knowledge belongs to traditional owners of the land. The present study found that Aboriginal people used about 78 medicinal plants against various inflammation and inflammatory-related conditions. Of the 78 species, 45 were already studied for their phytochemical and anti-inflammatory properties. However, most phytochemical, and pharmacological studies were reported from other countries, mainly China, India, Japan, Thailand, Vietnam, and Taiwan. Australia reported only five studies on *Alphitonia petriei*, *Angophora costata*, *Corymbia terminalis*, *Dodonaea polyandra*, and *Excoecaria agallocha*. Eighty-three isolated compounds have shown various anti-inflammatory activities in vitro assays (mostly LPS-activated macrophage cells) by inhibiting the production of pro-inflammatory cytokines and chemokines and the gene expression of mediators involved in the inflammatory pathways, including the NF-κB pathway.

Furthermore, those bioactive compounds in in vitro assays showed significant protection in the various animal disease models, mostly ear and paw oedema and colitis mice models. Most isolated bioactive compounds belonged to the chemical class of terpenes and diterpenoids, and flavonoids. Many Australian medicinal species are not yet studied and studying these plants will undoubtedly help the scientific validation and documentation of their traditional knowledge. It is evident from this review that many Aboriginal medicinal plants are associated with anti-inflammatory properties that can guide the discovery of novel anti-inflammatory drug leads in collaboration with the indigenous communities of Tropical Australia. 

## Figures and Tables

**Figure 1 molecules-27-03849-f001:**
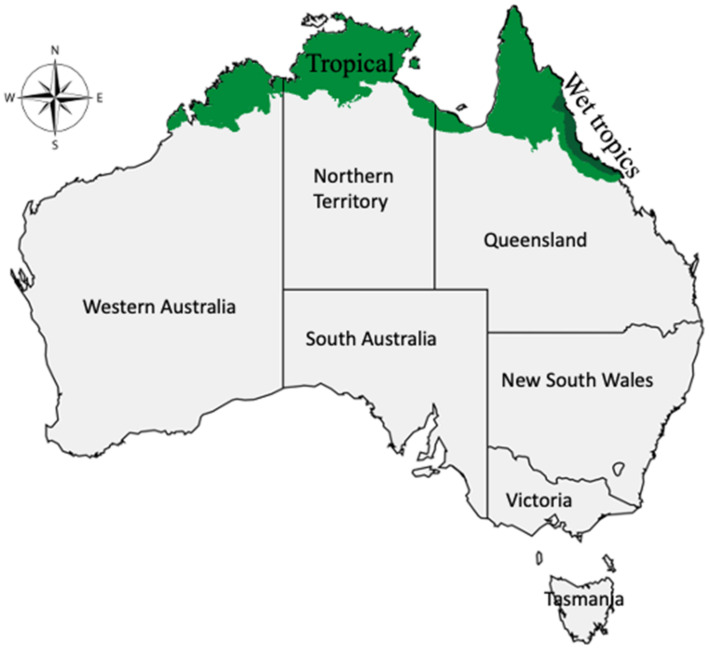
Map of Australia showing the tropical and wet tropics region. (Location labels and compass added; shaded ecoregions were hand-drawn using the information from an online climate map [20].

**Figure 2 molecules-27-03849-f002:**
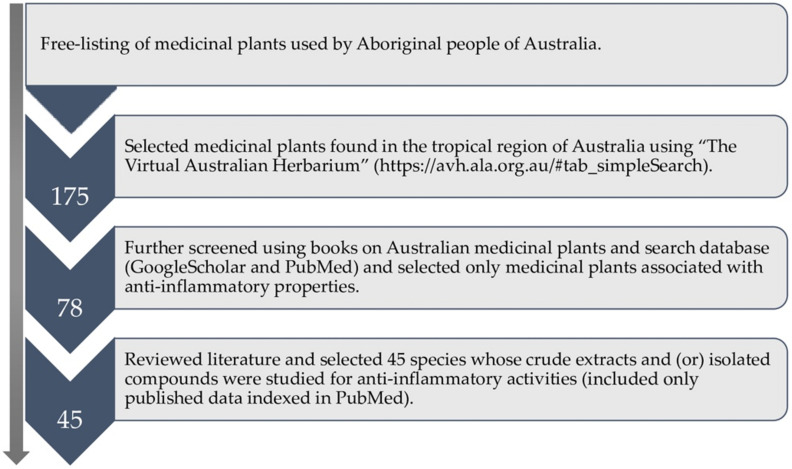
Number of anti-inflammatory medicinal plant species of tropical Australia selected against each criterion for their phytochemical and pharmacological properties review.

**Figure 3 molecules-27-03849-f003:**
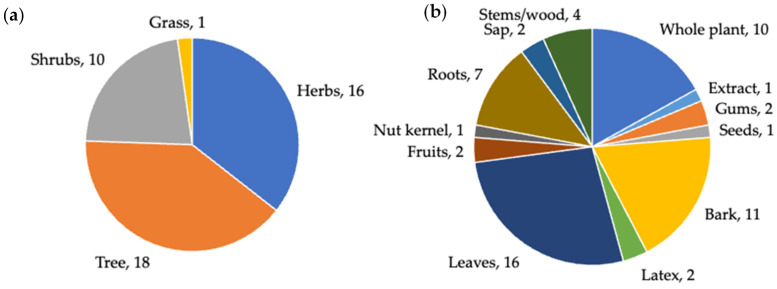
Anti-inflammatory medicinal plants of tropical Australia: (**a**) different life forms; (**b**) different parts used for medicinal purposes.

**Figure 5 molecules-27-03849-f005:**
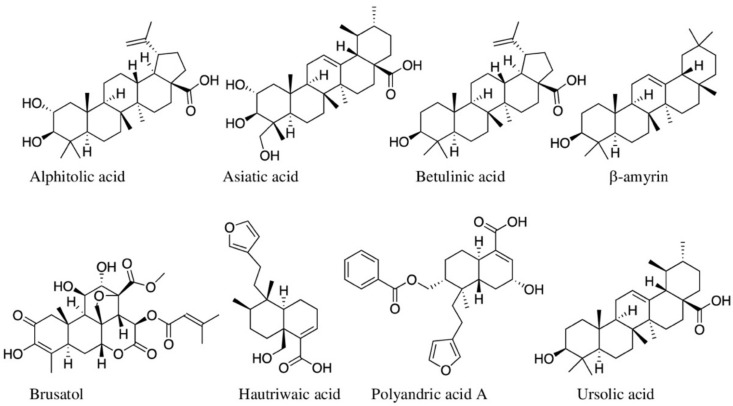
Chemical structures of anti-inflammatory terpenes and terpenoids.

**Figure 6 molecules-27-03849-f006:**
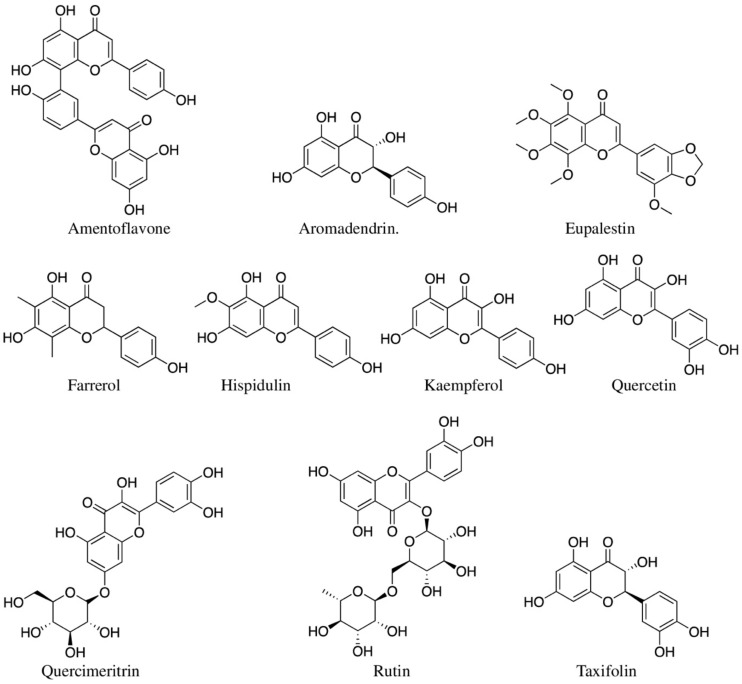
Chemical structures of anti-inflammatory flavonoids.

**Figure 7 molecules-27-03849-f007:**
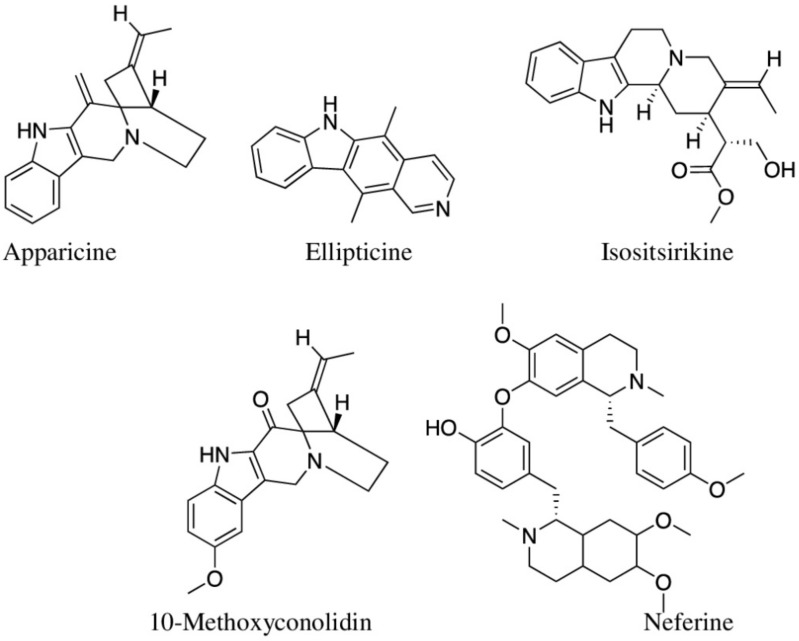
Chemical structures of anti-inflammatory alkaloids.

**Figure 8 molecules-27-03849-f008:**
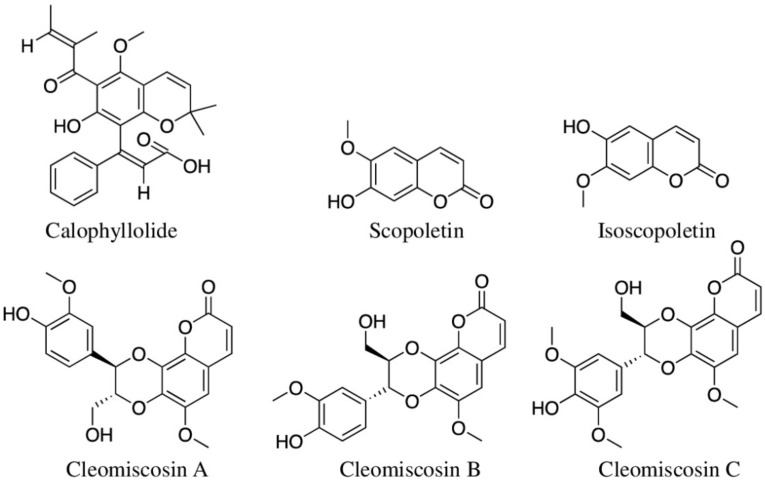
Chemical structure of anti-inflammatory coumarins.

**Figure 9 molecules-27-03849-f009:**
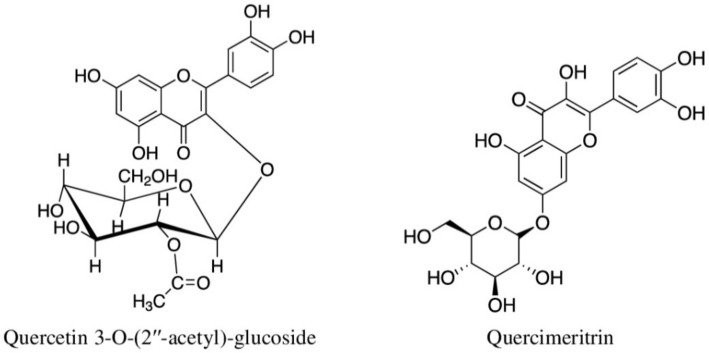
Chemical structures of anti-inflammatory glycosides.

## Data Availability

Not applicable.

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
