# Peer review of "Indigenous Uses, Phytochemical Analysis, and Anti-Inflammatory Properties of Australian Tropical Medicinal Plants"

_molecules, 2022, doi:10.3390/molecules27123849_

Round 1
Reviewer 1 Report
In this manuscript, the authors comprehensively reviewed the ethnopharmacology and phytochemistry, of the anti-inflammatory Australian tropical medicinal plants. Through consulting a large number of literatures, the anti-inflammatory medicinal plants were systematically summarized. I recommend this paper to be accepted after minor revision.
1. I suggest the authors to make modifications to the layout of Table 1. The information contained in the column of ethnomedical uses might be subdivided into ’preparation’ and ’application’. The ‘isolated compounds’ column might also be re-designed, as the large number of compounds made the information provided there not readable.
2. The subtitles in Section 5 were mis-labeled in present form. For example, Line 241, ‘5.4’ might be ‘5’; Line 462, ‘4.2.5’ might be ‘5.2.5’.
3. Line 292, ‘82’ might be ‘83’.
4. Section 5.2.3, additional information or discussion might be provided on the toxicity of alkaloids.
Author Response
We are very thankful for your positive comments and suggestions, and please see the attachment for details.

Reviewer 2 Report
The manuscript entitled “Anti-inflammatory Properties of the Australian Tropical Medicinal Plants: Evidences from Indigenous Uses, Phytochemical Analysis, and Pharmacological Evaluations” presents a significant work. But manuscript requires revision on basis of following points
1) Introduction section is not written well. There is need to elaborate pharmacological potential of plants and their importance in life of human beings To arouse a broad interest from readership in this field, some recent literature should be included i.e.,
https://www.sciencedirect.com/science/article/pii/S2214750014000493
https://doi.org/10.1016/j.arabjc.2013.04.024
https://link.springer.com/article/10.1186/s13065-018-0495-1
https://www.ajol.info/index.php/tjpr/article/view/178170
2) I am just curious to know, why authors have considered plants only on basis of anti-inflammatory
3) Generally, the manuscript needs extensive English improvements.
4) The title of this manuscript is not attractive, Please modify this ?
5) Please mention quantitative results of your studies in conclusion and abstract section.
6) Revised the abstract thoroughly
7) Table 5, 6, 7, 8 and 9: Please include the stereochemistry, if known, for all chiral centers.
8) Table 2 is the most informative aspect of the review, but it would benefit from additional clarification and editing. In addition to the antiinflammatories reported, the authors should critically assess the information. Do they represent “excellent”, “good”, “marginal”, “weak”, or “inactive”? The following paper https://www.mdpi.com/2218-273X/12/1/83 may be a good resource
9) - The reader should have, after the lecture of the paper, a summary of the representative compounds in investigated plants (not merely a list of all compound classes.) Which are marker compounds of antiiflammatory potent plants, if any? Are there any typical compounds in seeds/barks or leaves? Which compounds are quantitatively important?
10) There are numerous abbreviations throughout the text. These should be defined the first time they are used.
11) Reference section: Authors have added many self citations, which is really unethical and even citations are not placed at proper place ref # 4, 262, 280, 12 and 16. Self citations should be removed.
Author Response
We are incredibly thankful for providing valuable comments and suggestions. We have tried our best to incorporate all suggested changes to the manuscript. Please see the attachment for details.

Round 2
Reviewer 2 Report
Suggestions have been sufficiently incorporated. This paper is now considerable in present form